# CAN YOUR MODEL SEPARATE YOLKS WITH A WATER BOTTLE? BENCHMARKING PHYSICAL COMMONSENSE UNDERSTANDING IN VIDEO GENERATION MODELS

## ABSTRACT

Recent advances in text-to-video (T2V) generation have enabled visually compelling outputs, but models still struggle with everyday physical commonsense, often producing videos that violate intuitive expectations of causality, object behavior, and tool use. We introduce PhysVidBench, a human-validated benchmark for assessing physical reasoning in T2V models. It comprises carefully curated prompts spanning seven dimensions of physical interaction, from material transformation to temporal dynamics, offering broad, multi-faceted coverage of scenarios where physical plausibility is critical. For each prompt, we generate videos using diverse state-of-the-art models, and evaluate them through a three-stage pipeline: grounded physics questions are derived from each prompt, generated videos are captioned with a vision–language model, and a language model answers the questions using only the captions. This strategy mitigates hallucination and produces scores that align closely with human judgments. Beyond evaluation, PhysVidBench also serves as a diagnostic tool, enabling feedback-driven refinement of model outputs. By emphasizing affordances and tool-mediated actions, areas often overlooked in existing benchmarks, PhysVidBench provides a structured, interpretable framework for assessing and improving everyday physical commonsense in T2V models.

## 1 INTRODUCTION

Text-to-video (T2V) generation has recently made striking progress in visual quality, temporal coherence, and prompt alignment (Agarwal et al., 2025; Kong et al., 2024; Yang et al., 2025; Chen et al., 2024; Wang et al., 2025a; Kuaishou Team, 2024; Runway Team, 2024; Google DeepMind, 2025c). These models are increasingly positioned as *world video models* that could support robotics, embodied AI, and simulation-based learning (He et al., 2025; Hu et al., 2025; Wang et al., 2024; Hu et al., 2023; Liu et al., 2024c; Agarwal et al., 2025), where understanding physical interactions is crucial. Yet, despite impressive realism, current models frequently violate *everyday physical commonsense*: objects float unnaturally, tools are misused, and causal sequences break down.

Several recent benchmarks have aimed to quantify physical reasoning in video generation. Phy-GenBench (Meng et al., 2024) tests isolated laws such as buoyancy and friction, which limits its applicability to more complex interactions. VideoPhy 2 (Bansal et al., 2025) focuses on motion plausibility and physics violations but omits affordances and tool use, while relying on vision-language model (VLM) scoring that is prone to hallucinations. VBench-2.0 (Zheng et al., 2025) evaluates temporal consistency and state changes but depends on predefined reasoning classes and multi-question VLM pipelines that may lack strong grounding. While valuable, these efforts leave gaps in assessing richer, tool-mediated interactions and intuitive physical understanding. To address this, we introduce **PhysVidBench**, a benchmark designed to assess everyday physical reasoning in T2V models through real-world tasks that require understanding of object functionality and tool-mediated interactions. A structured comparison with prior benchmarks is shown in Table 1.

PhysVidBench comprises 383 prompts adapted from the PIQA dataset (Bisk et al., 2019), which focuses on everyday physical scenarios. Unlike synthetic setups, our prompts reflect routine activities such as manipulating tools, transferring materials, or executing household tasks. Figure 1 illustrates

Table 1: **Comparison of PhysVidBench with prior physical commonsense video benchmarks.** PhysVidBench uniquely emphasizes tool use and affordances, introduces upsampled prompts for causal specificity, and employs a caption-based Yes/No QA evaluator aligned with human judgments.

| Feature | PhyGenBench (Meng et al., 2024) | Vbench-2.0 (Zheng et al., 2025) | VideoPhy-2 (Bansal et al., 2025) | PhysVidBench (Ours) |
|---|---|---|---|---|
| # of Prompts | 160 | 1013 | 3940 | 383 |
| Tool & Affordance Focus | ✗ | ✗ | ✗ | ✓ |
| Human Alignment Study | ✓ | ✓ | ✓ | ✓ |
| Upsampled Prompt Comparison | ✓ | ✗ | ✓ | ✓ |
| Hard Subset | ✗ | ✗ | ✓ | ✓ |
| Automatic Evaluation | ✓ | ✓ | ✓ | ✓ |
| Human Feedback Type | Rating | Pairwise | Rating (0-1) | Yes/No QA |

**Real-World Videos**  **Synthetically Generated Videos**

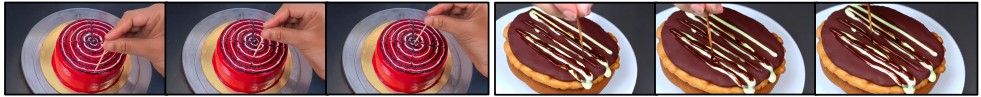

Hands squeeze an empty plastic water bottle, place the opening over an egg yolk in a bowl, and release the squeeze, sucking the yolk into the bottle, leaving the white behind.

**Tested concepts:** fundamental physics (⚙), object properties & affordances (▱), spatial reasoning (⬦), action & procedural understanding (⛓) force and motion (💪)

A toothpick drags through drizzled sauce on top of frosting, creating wavy lines.

**Tested concepts:** object properties & affordances (▱), spatial reasoning (⬦), material interaction & transformation (🔺), force and motion (💪)

Figure 1: **Understanding Physical Commonsense in Video Generation Models.** Humans intuitively grasp how everyday interactions unfold; how objects respond to forces, how tools function, and how materials behave under manipulation. To test whether T2V models have this capability, we present *PhysVidBench*, a benchmark for evaluating key dimensions of physical commonsense. The figure shows two examples comparing real-world and model-generated videos: one involving suction via a squeezed bottle to separate an egg yolk, and another showing a toothpick dragging sauce across frosting to create patterns. Each case tests multiple concepts, such as fundamental physics, object affordances, spatial reasoning, and material transformations. Despite visual realism, current models often violate basic physical expectations, revealing persistent gaps in physical reasoning.

two representative scenarios drawn from the real-world physical interactions underlying our prompts, shown side-by-side with their model-generated counterparts. This contrast highlights how simple, everyday physical interactions that humans perform remain challenging for state-of-the-art T2V models. Each prompt has an upsampled variant that enriches causal and material detail, stressing model understanding. The benchmark is structured around seven core reasoning dimensions, including force and motion, object affordance, spatial reasoning, and material interaction (Fig. 2).

To evaluate generated videos, we introduce a caption-based question-answering pipeline. From each prompt, grounded yes/no physics questions are derived; videos are captioned by a VLM; and a large language model (LLM) answers the questions using only the captions. This design avoids pitfalls of direct VLM scoring, which are prone to hallucination and often lack grounded physical understanding (Huang et al., 2024a; Chow et al., 2025). All generated questions are manually validated, and the resulting evaluator aligns strongly with human judgments, substantially outperforming direct VLM baselines as demonstrated in our experiments. Beyond evaluation, PhysVidBench provides diagnostic feedback. Following the iterative self-refinement paradigm of PhyT2V (Xue et al., 2025a), our benchmark supports a lightweight refinement loop: models guided by PhysVidBench improve their accuracy over baseline outputs, without retraining.

Our contributions include: (i) PhysVidBench, a new benchmark and dataset derived from 383 PIQA prompts, designed to evaluate physical commonsense in T2V models with a focus on tool use and object affordance; (ii) a modular and interpretable evaluation pipeline that avoids direct VLM-based reasoning and leverages caption-based LLM judgments for physical commonsense; and

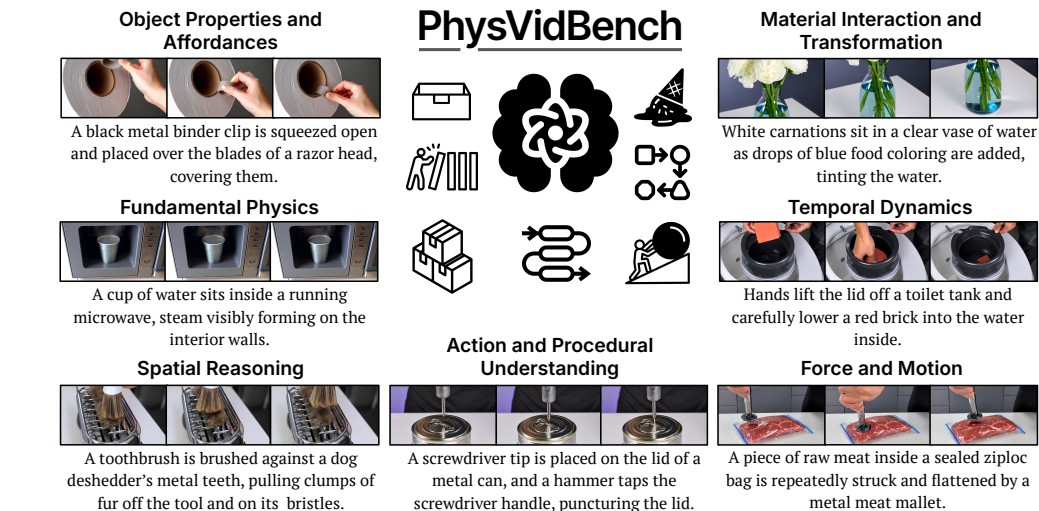

Figure 2: **Physical commonsense dimensions tested in PhysVidBench**, each illustrated with a video generated by Cosmos-14B. Prompts are designed to probe specific categories such as force and motion, object affordance, spatial containment, temporal progression, and material interaction. Each row shows sampled frames from one generated video paired with its corresponding prompt.

(iii) an empirical analysis across a wide range of T2V models that reveals persistent gaps in physical reasoning capabilities, while demonstrating PhysVidBench's utility as a diagnostic tool for guiding model refinement. We release the dataset and evaluation code.

## 2    RELATED WORK

**Video Generation Models.** Recent years have seen rapid advances in text-to-video (T2V) generation. Models such as CogVideoX (Yang et al., 2025), HunyuanVideo (Kong et al., 2024), and VideoCrafter2 (Chen et al., 2024) demonstrate strong temporal coherence and visual quality. MAGVIT (Yu et al., 2023) explores masked video transformers, while large-scale systems like Cosmos (Agarwal et al., 2025) and Wan (Wang et al., 2025a) focus on scalability and prompt alignment. Proprietary models such as Sora (Runway Team, 2024) and Veo-2 (Google DeepMind, 2025c) further push the limits of video fidelity and duration. Recent work also draws on world models that aim to encode structured physical and causal dynamics. While visual realism has improved, whether these systems possess a grounded understanding of physical interactions remains an open challenge, one that our work directly addresses.

**Metrics and Benchmarks for Video Generation.** Early evaluation of T2V models relied on image-based metrics like CLIP Score (Hessel et al., 2021), Fréchet Video Distance (FVD) (Unterthiner et al., 2018), and Inception Score (IS) (Salimans et al., 2016). These metrics capture alignment, diversity, or distributional similarity, but overlook temporal consistency, motion plausibility, and physical validity, often misaligning with human judgment (Huang et al., 2024b). More recent benchmarks improve granularity: VBench (Huang et al., 2024b) measures motion consistency and spatial stability, EvalCrafter (Liu et al., 2024b) evaluates visual, content, and motion quality, and FETV (Liu et al., 2023) organizes prompts by content and complexity to better match human ratings. Yet, these benchmarks remain limited in their treatment of physical commonsense, motivating the need for dedicated evaluations of real-world plausibility.

**Benchmarks for Physical Commonsense**. Several datasets explicitly probe physical reasoning in video models. VBench-2.0 (Zheng et al., 2025) introduces dimensions such as state change, geometry, and motion rationality to assess consistency and transformations. VideoPhy (Bansal et al., 2024) and VideoPhy-2 (Bansal et al., 2025) adopt an action-centric approach, focusing on whether videos of real-world activities adhere to physical commonsense. PhyGenBench (Meng et al., 2024) emphasizes isolated physical laws such as buoyancy and friction, making it less applicable to complex tool-mediated interactions. Physics-IQ (Motamed et al., 2025) evaluates generative models on fluid,

optical, and electromagnetic phenomena, while Cosmos-Reason1 (Agarwal et al., 2025) benchmarks reasoning across fundamental domains like space, time, and fundamental physics. Although these efforts advance physical evaluation, they largely omit tool use and affordance reasoning, and rely on predefined categories that may fail to capture the multi-faceted nature of everyday interactions. PhysVidBench fills this gap by focusing on *everyday physical commonsense*, with prompts centered on tool-mediated interactions and a human-aligned evaluation pipeline.

**Improving Physical Consistency in Video Generation.** Several recent works explicitly aim to improve the physical plausibility of generated videos. PhyT2V (Xue et al., 2025a) uses an LLM-guided self-refinement loop that rewrites prompts based on detected violations, while WISA (Wang et al., 2025b) injects structured physical knowledge through a world-simulator assistant and specialized attention mechanisms. VideoREPA (Zhang et al., 2025) aligns T2V diffusion models with video-understanding foundation models by distilling spatio-temporal relational structure. Complementary approaches incorporate physics constraints directly during training: PhysDiff (Yuan et al., 2023) enforces Newtonian motion consistency in diffusion-based video prediction, and PhysGen (Liu et al., 2024a) supervises generative models using differentiable physics rollouts to encourage realistic trajectories and contacts. While these methods aim to improve physical consistency, PhysVidBench offers a model-agnostic, human-aligned benchmark that tests whether these physics-aware techniques genuinely improve everyday physical commonsense in tool-use and affordance-driven scenarios.

**Physical Reasoning in LLMs and VLMs.** Physical commonsense also remains a challenge for LLMs and VLMs. PIQA dataset (Bisk et al., 2019) was introduced to assess models' understanding of everyday physical interactions, such as selecting appropriate tools for tasks. It consists of thousands of human-authored commonsense questions framed as short goals, each paired with one correct and one incorrect solution. While humans achieve near-perfect accuracy on PIQA, LLMs still lag behind, highlighting their limited grasp of physical commonsense. Similarly, PhysBench (Chow et al., 2025) evaluates VLMs across tasks involving object properties, relationships, scene understanding, and physics-driven dynamics, finding that most fail to capture physical world knowledge. Impossible Videos (Bai et al., 2025b) tests video models with scenarios that defy physical laws, with findings indicating that they often fail to recognize physically implausible events.

*(See Appendix B for a detailed comparison of PhysVidBench against prior physical-commonsense video benchmarks.)*

## 3 BENCHMARK

### 3.1 PHYSICAL COMMONSENSE CONCEPTS

Evaluating whether T2V models truly understand the physical world requires more than judging visual fidelity or motion smoothness. It demands a systematic framework that decomposes physical reasoning into interpretable dimensions, reflecting how humans interact with and reason about their environment. We adopt a top-down methodology grounded in everyday human experience. Instead of abstract physics categories or synthetic scenes, we start from goal-solution pairs in the PIQA dataset (Bisk et al., 2019), which capture real-world, physically plausible tasks such as scraping ice with a credit card or folding materials to fit a container. These naturally occurring scenarios require implicit understanding of force, material behavior, and procedural logic, making PIQA a rich source for constructing physically grounded prompts. From these tasks, we derive a data-driven ontology of physical commonsense. Each dimension captures a distinct reasoning capability, ranging from force and motion to procedural understanding. Table 2 summarizes the seven dimensions, which structure PhysVidBench to evaluate not just visual quality, but whether models grasp the principles underlying everyday physical interactions.

### 3.2 PROMPT CONSTRUCTION

PhysVidBench is designed to evaluate whether T2V models can depict physically plausible everyday scenarios. Each prompt describes a short, concrete action that can be directly visualized and reasoned about. The construction pipeline proceeds in three stages (Fig. 3): (1) selection and filtering of physically plausible goal-solution pairs, (2) generation of base video prompts, and (3) enrichment of base prompts to highlight underlying physical properties and causal structures.

Table 2: **Definitions of physical reasoning categories in PhysVidBench.** These seven dimensions form the core of our PhysVidBench benchmark and are derived from real-world scenarios. The seven categories cover complementary aspects of everyday commonsense reasoning and enable fine-grained evaluation of T2V models. The taxonomy supports fine-grained evaluation of text-to-video models along dimensions such as causality, spatial relations and material properties.

| Commonsense Dimension | Definition |
| --- | --- |
| Fundamental Physics | Core physical principles such as energy conservation, causality, equilibrium, and state transitions in physical systems. |
| Object Properties & Affordances | Inherent object attributes including material composition, rigidity, softness, and functional affordances like containment or support. |
| Spatial Reasoning | Spatial relations including position, geometry, occlusion, orientation, and fit between objects in a scene. |
| Temporal Dynamics | Timing, sequencing, and the causal structure of events over time, such as ordering, delays, and waiting. |
| Action & Procedural Understanding | Structured, goal-driven sequences of actions involving procedural steps toward task completion. |
| Material Interaction & Transformation | Responses of materials to external forces or processes including transformations like melting, freezing, breaking, or chemical change. |
| Force and Motion | Physical interactions governed by forces, including motion, pushing, pulling, lifting, and properties like inertia. |

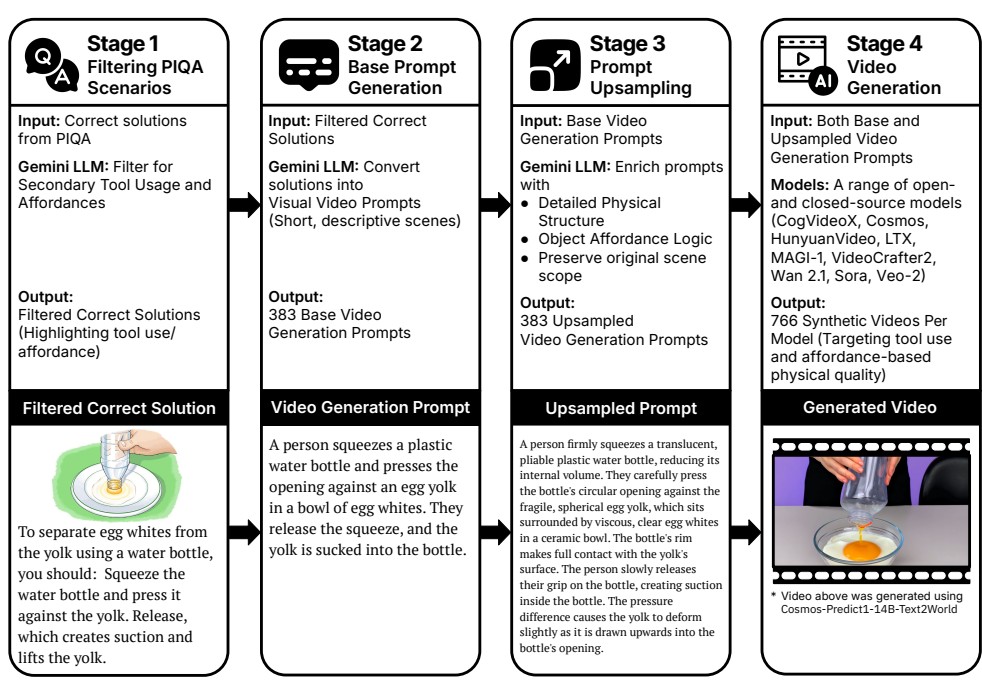

Figure 3: **Video generation pipeline for PhysVidBench.** From PIQA goal-solution pairs, we filter instances for secondary tool use and object affordances (Stage 1). These are converted into short, physically grounded prompts (Stage 2), which are then enriched with additional details (Stage 3). Base and upsampled prompts are used to generate videos across multiple T2V models (Stage 4).

**Stage 1: Filtering PIQA Scenarios.** PIQA offers thousands of human-written goal–solution pairs capturing routine, physically grounded activities such as moving objects, manipulating tools, preparing food, and handling materials. These pairs were crowd-sourced under instructions to propose common, physically plausible tasks from everyday life, resulting in broad coverage of forces, materials, tools, and object–object interactions. This makes PIQA a diverse and realistic foundation

for constructing physically meaningful video prompts. We extract correct goal–solution pairs from the PIQA training and validation splits, ensuring that all candidates are physically viable. To emphasize challenging cases, we apply a lightweight filtering step using Gemini-2.5-Pro-Preview-05-06 (Google DeepMind, 2025a) (hereafter referred to as Gemini 2.5 Pro). Specifically, we retain goal-solution pairs from PIQA that involve *secondary tool use* or *non-obvious object affordances*, such as using a credit card to scrape ice or a paperclip to eject a SIM tray. These scenarios go beyond standard object usage and demand an intuitive understanding of how tools can be repurposed based on their physical properties. This filtering sets our benchmark apart from prior efforts, which typically rely on canonical actions or predefined physics categories.

**Stage 2: Base Prompt Generation.** Each filtered pair is converted into a concise video prompt using Gemini 2.5 Pro. Prompts are required to depict realistic, self-contained short video clips centered on observable physical interactions such as placing, cutting, lifting, folding, or pouring. To ensure clarity and physical grounding, the model is explicitly instructed to exclude non-visual elements such as internal states, abstract concepts, or speculative intentions, and to discard input pairs that cannot be translated into meaningful, demonstrable actions. This process yields 383 base prompts, each describing a physically grounded real-world scenario suitable for text-to-video generation. The resulting prompt set shows substantially richer physical variation than prior physical-commonsense benchmarks. Across the 383 prompts, PhysVidBench spans more than 1,200 distinct objects and over 600 manipulation actions, covering a wide range of materials, tools, and interaction types encountered in everyday settings. PhysVidBench surpasses PhyGenBench, VBench-2.0, and VideoPhy-2 in both lexical richness and action–object interaction density, showing that the dataset is diverse across real-world object usage and physical interactions.

*(See Appendix B.2 for quantitative diversity statistics.)*

**Stage 3: Prompt Upsampling.** Each base prompt is further refined through an upsampling step using Gemini 2.5 Pro, guided by instructions that expand on the physical structure and affordance logic of the scene. This process, inspired by recent work such as PhyT2V (Xue et al., 2025a), enriches prompts with object-level attributes that enhance realism and physical plausibility in T2V outputs. Crucially, the upsampling is constrained to grounded enhancements: the model is instructed to preserve the original scene scope and avoid introducing new objects, events, or speculative consequences. The result is an enriched prompt that retains the core action of its base version while making its physical details and causal logic more explicit. This procedure yields 383 upsampled prompts, producing a total of 766 unique prompts (383 base + 383 upsampled).

All base and upsampled prompts were manually reviewed by human annotators to remove hallucinations, unsafe or inappropriate content, and to ensure suitability for video generation. Videos were then generated for every prompt using each evaluated text-to-video model, allowing us to assess not only surface-level fidelity but also the ability to follow nuanced, causally rich, and physically grounded instructions. The complete generation pipeline, from filtering to prompt enrichment and final model outputs, is summarized in Fig. 3.

*(See Appendix C.1 and C.2 for full details of the prompt construction process and LLM instructions.)*

### 3.3 EVALUATION PIPELINE

We adopt a three-stage pipeline shown in Fig. 4 to assess physical commonsense in generated videos.

**Stage 1: Physics-Grounded Question Generation.** From each upsampled prompt, we derive targeted yes/no questions whose correct answer is always yes because each prompt specifies the physical phenomenon that should occur in a correct video. These questions are designed to be answerable solely from the prompt itself, without requiring any external knowledge. Using only yes-grounded questions avoids the ambiguity of no-questions, where many physically incorrect outcomes collapse to the same answer and fail to indicate whether the intended behavior actually happened. For example, if a prompt expects an object to fall, a "No" question such as "Does the object stay suspended?" would treat a wide range of incorrect generations, falling through the floor, disappearing, melting, or teleporting, as equally valid answers, even though these reflect fundamentally different and physically implausible errors. In our setup, a model is scored as correct only when the generated video provides positive evidence for the expected physical effect; physically implausible or failed generations naturally receive a "no," producing a stable and interpretable

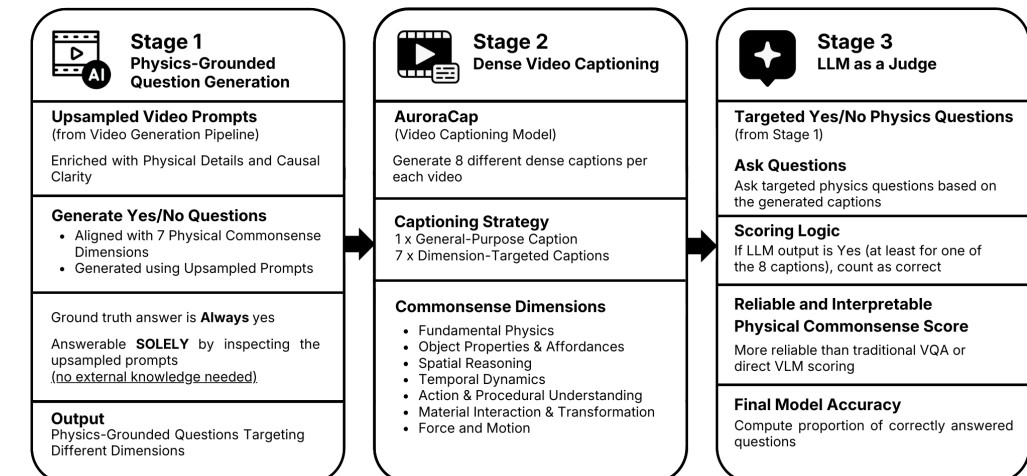

Figure 4: **Evaluation pipeline for PhysVidBench.** From each upsampled prompt, we generate targeted yes/no questions aligned with one or more physical commonsense dimensions (Stage 1). For every generated video, AuroraCap produces a general-purpose caption along with seven dimension-specific captions that highlight relevant physical cues (Stage 2). These captions, together with the questions, are then provided to an LLM, which answers based solely on the textual evidence (Stage 3).

measure of physical commonsense. This process yields 4,123 questions, averaging 11 per video. All questions were manually reviewed for clarity, answerability, and appropriateness; about 5% were discarded for referencing non-visual elements (e.g., sounds) or details present only in the upsampled prompt but absent from the corresponding base prompt.

**Stage 2: Dense Video Captioning.** Each generated video is passed to a dense captioning model that produces one general-purpose caption plus seven dimension-specific captions aligned with our ontology. In our main experiments, we use AuroraCap (Chai et al., 2025)[1]. The dimension-specific captions emphasize details relevant to their category, such as object properties, material behavior, or spatial relationships. Collectively, the eight captions provide complementary perspectives on the video, improving coverage of fine-grained physical features and reducing dependence on any single potentially incomplete description, addressing key challenges in VLM-based evaluation (Huang et al., 2024a; Chow et al., 2025).

**Stage 3: LLM as a Judge.** We use a large language model to answer each question using only the eight generated captions without access to the video or the original prompt. Because this evaluation is fully text-based, the judge does not observe the raw video but only its captioned description, enabling a modular separation between perception (captioning) and reasoning (LLM QA). In our main experiments, we use Gemini-2.5-Flash (Google DeepMind, 2025b)[2]. A response is marked correct if the model answers `yes` for at least one caption. Because the LLM must justify a "yes" using evidence from the captions, a T2V model cannot obtain a correct score unless the expected physical behavior actually appears in the video, as physically incorrect videos simply lack the textual cues required for a yes-answer and are therefore penalized. Final accuracy is computed as the proportion of correctly answered questions using an LLM-as-judge approach, following recent trends in leveraging language models for grounded evaluation (Zheng et al., 2023).

*(See Appendices C.3, E, H.2, and H.3 for evaluation details, an end-to-end example, cross-judge LLM analysis, and captioner robustness.)*

---

[1]We find that replacing it with captioners such as Qwen-2.5-VL-7B (Bai et al., 2025a) or Qwen-3-VL-30B (Team, 2025) produces comparable accuracies and model rankings, indicating that the pipeline is modular and compatible with different high-quality video captioners.

[2]We obtain nearly identical model rankings when replacing it with DeepSeek, indicating that the evaluation stage is not dependent on a specific LLM.

## 4  EXPERIMENTAL SETUP

**Models.** We evaluate a range of recent open and closed-source text-to-video (T2V) models covering diverse architectures and training paradigms. These include: (1) *VideoCrafter2* (Chen et al., 2024), a diffusion-based model for efficient video synthesis; (2) *CogVideoX-2B/5B* (Yang et al., 2025), autoregressive models for high-fidelity generation; (3) *Wan2.1-1.3B/14B* (Wang et al., 2025a), trained on large-scale multimodal corpora; (4) *Cosmos-7B/14B* (Agarwal et al., 2025), instruction-tuned for complex generation tasks; (5) *HunyuanVideo* (Kong et al., 2024), an open-source release from Tencent's Hunyuan framework; (6) *LTX-Video* (HaCohen et al., 2024), a diffusion model using latent temporal encoding; (7) *MAGI-1* (Sand-AI, 2025), a motion-aware autoregressive model; (8) *Sora* (OpenAI, 2025) and (9) *Veo-2* (Google DeepMind, 2025c) are state-of-the-art proprietary models for high-quality video generation. Our benchmark includes both open-source and proprietary models to provide a broader view of current capabilities. All evaluations were conducted on a machine with a single NVIDIA A40 GPU.

**Difficulty–Based Prompt Stratification.** To better characterize model capabilities across varying levels of complexity, we partitioned our benchmark into three difficulty-based subsets: *medium*, *hard*, and *very hard*. We emphasize that even the medium subset poses significant challenges for current models, hence our terminology does not include an easy category. Our stratification process was as follows. First, we selected the four best-performing open-source models on the full benchmark: Cosmos, Hunyuan, MAGI-1, and Wan2.1. For each prompt, we computed the average question-answering accuracy across these models. Prompts were categorized according to the aggregated scores: medium if the average accuracy exceeded 40%, hard if it fell between 20% and 40%, and very hard if it was below 20%. This process yielded 123 medium, 160 hard, and 100 very hard prompts. We release the medium and hard subsets publicly, reserving the very hard set for future evaluations.

## 5  RESULTS AND DISCUSSION

**Overall Model Comparison.** Table 3 reports model accuracies across the seven ontology dimensions in PhysVidBench. To provide a holistic view, Figure 5 compares models using violin plots of accuracy distributions. Open-source systems such as Cosmos-14B and Wan2.1-14B achieve the highest averages but still show wide variance across prompts. Proprietary models (Sora, Veo-2) perform competitively on some dimensions yet fail in others, reinforcing that no current system consistently handles the full range of everyday physical commonsense. Table 4 presents model accuracies across the three difficulty levels using upsampled prompts, again with parentheses indicating the change relative to base prompts. Performance decreases sharply as difficulty increases: most models lose more than half their accuracy when moving from medium to very hard subsets.

*(See Appendix F.1 for qualitative examples across models.).*

**Dimension-wise Performance.** Two consistent trends are observed. First, *Spatial Reasoning (SR)* and *Temporal Dynamics (TD)* remain the most challenging, which require reasoning over geometry, occlusion, and event sequencing. Even the strongest models such as such as Wan2.1-14B, Cosmos-14B, Sora, and Veo-2 underperform on these dimensions, highlighting their persistent limitations in relational and temporal modeling. Second, *Object Properties & Affordances (OP)* yields the highest scores, suggesting that models often succeed by exploiting simple visual heuristics (e.g., texture, containment) rather than robust causal reasoning. Although results vary across architectures, overall scores remain modest, indicating that despite impressive visual fidelity, current T2V models lack generalizable physical commonsense.

*(See Appendix D for a structured taxonomy of prompt-level difficulty, which characterizes why certain prompts are inherently harder for current T2V models.)*

**Effect of Prompt Enrichment.** We next examine the role of prompt enrichment by comparing base and upsampled prompts. Across nearly all models, upsampled prompts improve accuracy, surfacing latent physical knowledge when fine-grained attributes are made explicit. Interestingly, proprietary models behave differently: Sora performs worse with enriched prompts, likely due to internal prompt-expansion mechanisms. These findings confirm that enrichment is broadly beneficial, though its effectiveness depends strongly on model architecture and training strategy.

Table 3: **Model performance across physical commonsense dimensions.** Reported values indicate accuracy with upsampled prompts; values in parentheses show the change when models switch to base prompts. Evaluation covers seven key physical commonsense dimensions as defined in our ontology, **AU**: Action & Procedural Understanding, **FM**: Force and Motion, **FP**: Fundamental Physics, **MT**: Material Interaction & Transformation, **OP**: Object Properties & Affordances, **SR**: Spatial Reasoning, **TD**: Temporal Dynamics. Bold and underline denotes the best and second best methods.

| Model | AU | FM | FP | MT | OP | SR | TD | Average |
|---|---|---|---|---|---|---|---|---|
| LTX-Video | 20.7 (-4.2) | 19.9 (-3.5) | 18.6 (-2.5) | 16.9 (-1.6) | 25.0 (-4.7) | 17.3 (-2.3) | 13.7 (-4.6) | 21.0 (-3.0) |
| VideoCrafter2 | 17.7 (-1.4) | 19.4 (-2.9) | 23.3 (-3.3) | 18.8 (-2.9) | 26.6 (-2.1) | 15.8 (-0.2) | 12.6 (-2.7) | 22.1 (-1.7) |
| CogVideoX (2B) | 22.6 (-1.8) | 24.6 (-4.5) | 23.8 (-2.7) | 22.6 (-0.3) | 28.9 (-3.5) | 19.6 (-1.3) | 16.8 (-1.5) | 25.6 (-3.1) |
| CogVideoX (5B) | 18.9 (-0.8) | 19.6 (-3.3) | 21.3 (-3.2) | 18.1 (-1.4) | 25.6 (-1.3) | 17.9 (-0.5) | 12.6 (+0.8) | 21.0 (-1.5) |
| Wan2.1 (1.3B) | 30.1 (-3.5) | 29.0 (-5.0) | 28.5 (-2.8) | 28.9 (-1.9) | 35.2 (-3.6) | 27.4 (-3.1) | 18.7 (+0.0) | 30.6 (-3.1) |
| Wan2.1 (14B) | 33.6 (-3.5) | 31.6 (-1.4) | 32.6 (-1.3) | 32.9 (-3.5) | 39.0 (-3.9) | 29.6 (-1.3) | **23.7** (-1.5) | 34.1 (-1.9) |
| MAGI-1 | 27.3 (-5.9) | 26.6 (-4.0) | 29.4 (-1.7) | 28.5 (-5.1) | 37.4 (-5.0) | 30.4 (-5.0) | 19.1 (-8.0) | 32.6 (-5.3) |
| Hunyuan Video | 26.7 (-0.5) | 26.2 (+0.2) | 28.1 (-0.1) | 32.3 (-4.8) | 36.1 (-1.9) | 23.9 (+2.1) | 21.8 (-5.3) | 30.0 (-0.9) |
| Cosmos (7B) | 32.5 (-8.2) | 32.8 (-9.3) | **33.6** (-8.9) | 36.1 (-10.6) | 40.1 (-10.6) | 31.6 (-10.0) | 21.4 (-8.4) | 35.0 (-8.3) |
| Cosmos (14B) | **35.6** (-5.9) | 33.3 (-6.1) | 31.7 (-3.9) | **37.4** (-5.9) | **40.9** (-9.6) | **32.5** (-5.3) | 21.0 (-0.4) | **36.2** (-7.1) |
| Sora | 28.6 (+2.4) | 30.0 (-1.1) | 29.9 (+0.2) | 33.1 (+0.3) | 37.0 (-1.1) | 24.8 (+1.7) | 16.7 (+2.6) | 31.4 (+0.3) |
| Veo-2 | 34.9 (-0.7) | **33.8** (-2.3) | 31.7 (-2.7) | 36.0 (-3.1) | 38.4 (-1.5) | 29.7 (+1.0) | 19.5 (+2.5) | 34.8 (-0.7) |

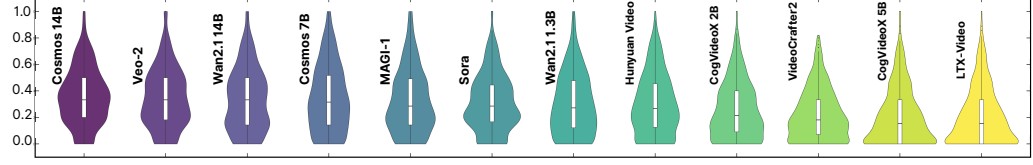

Figure 5: **Performance comparison of video generation models on PhysVidBench.** Each violin plot represents a distinct model and depicts the distribution of accuracy scores across all prompts.

Table 4: **Model performances on difficulty based subsets.** Reported values indicate accuracy with upsampled prompts; values in parentheses show the change when models switch to base prompts.

| Model | Medium | Hard | Very Hard |
|---|---|---|---|
| LTX-Video | 41.2 (-1.6) | 22.8 (-4.4) | 9.1 (-1.2) |
| VideoCrafter2 | 44.5 (-4.3) | 23.1 (-1.1) | 10.7 (-1.6) |
| CogVideoX (2B) | 49.8 (-7.8) | 28.2 (-3.2) | 11.6 (-0.6) |
| CogVideoX (5B) | 39.2 (-2.7) | 22.0 (-0.7) | 11.3 (-1.1) |
| Wan2.1 (1.3B) | 57.4 (-3.8) | 33.8 (-4.7) | 14.6 (-1.3) |
| Wan2.1 (14B) | 61.5 (-4.8) | 37.0 (-2.7) | 18.0 (+0.2) |
| MAGI-1 | 60.5 (-10.7) | 35.2 (-5.5) | 16.8 (-3.2) |
| Hunyuan Video | 52.6 (-4.5) | 33.5 (-5.1) | 15.9 (-4.8) |
| Cosmos (7B) | 60.5 (-10.7) | 38.8 (-10.3) | 18.6 (-5.0) |
| Cosmos (14B) | 59.2 (-13.1) | 39.2 (-8.7) | 22.2 (-2.2) |

**Impact of Model Scale.** Larger models generally achieve higher scores, but scaling alone does not guarantee stronger physical reasoning. Wan2.1-14B consistently outperforms its 1.3B variant, and Cosmos-7B exceeds smaller counterparts, yet CogVideoX-2B outperforms CogVideoX-5B across all categories. Moreover, scaling gains plateau: even the largest models struggle with SR and TD. Notably, several smaller models such as MAGI-1(4.5B) and Wan2.1-1.3B outperform larger systems like VideoCrafter2(2B) and CogVideoX-5B, indicating that architectural choices, training data, and intermediate supervision often matter more than parameter count. Taken together, these findings suggest that scaling helps models exploit explicit physical cues, but does not resolve deeper deficiencies in temporal or relational reasoning.

**Validation of our Evaluation Pipeline.** To assess the reliability of our evaluation pipeline, we conducted a user study with 15 participants and 120 videos. Participants answered randomized yes/no questions about the generated videos, enabling a direct comparison between human responses and automatic scores. We benchmarked our method against two alternatives: (i) direct VLM-based QA, where the model answers physics questions directly from the video, and (ii) two public automatic metrics designed for different objectives (aesthetics/overall quality and dynamics realism). As shown in Table 5, our evaluation pipeline achieves substantially stronger alignment with human judgments. In contrast, VLMs often underperform due to response collapse or reliance on superficial cues, while existing metrics show weak or even negative correlation with human ratings.

*(See Appendix G for details of the user study and Appendix H for additional robustness analyses.)*

Table 5: **Comparison of direct use of VLMs, prior evaluation metrics and our evaluation pipeline.** Reported values are Pearson correlations with human responses on 120 videos. "N/A" denotes failure cases where a VLM collapsed (e.g., answering `yes` to nearly all questions), preventing meaningful correlation. Our pipeline achieves consistently stronger alignment with human ratings than both direct VLM-based evaluation and existing automatic metrics.

| Method | MAGI-1 | Cosmos-14B | Hunyuan | Wan2.1 (14B) |
|---|---|---|---|---|
| Qwen2.5-VL-72B-Instruct-AWQ | 0.47 | 0.36 | 0.42 | 0.21 |
| Video-LLaVA | N/A | N/A | N/A | N/A |
| InternVL3 | N/A | N/A | N/A | N/A |
| VideoScore | -0.26 | 0.24 | -0.09 | -0.08 |
| VideoPhy-2 AutoEval | 0.23 | -0.18 | 0.16 | -0.06 |
| Ours (Captioning + LLM-as-a-Judge) | **0.69** | **0.60** | **0.61** | **0.45** |

**Iterative Error-Guided Prompt Refinement.** Inspired by the self-refinement paradigm in (Xue et al., 2025b), we extend our evaluation pipeline into an *iterative, training-free* development loop. Each round begins with a base prompt $p_t$, generates a video $v_t$, and applies our QA pipeline across the seven physical commonsense dimensions to obtain binary outcomes per question. *Violations*, questions answered incorrectly, are then used to condition the next prompt $p_{t+1}$ through targeted edits, such as adding object attributes, refining interaction verbs, inserting temporal cues, or applying explicit negatives. This process is repeated until performance on a held-out subset plateaus or a maximum number of rounds is reached. Unlike reward optimization or policy-gradient approaches, this procedure does not update model parameters. Instead, it re-edits prompts with failure-aware constraints while preserving generative diversity. On a 100-prompt subset, the approach produces substantial improvements: CogVideoX–2B rises from 21.6 to 32.7, and CogVideoX–5B from 17.8 to 29.7. These results demonstrate that our pipeline used for assessment can also guide effective refinement, highlighting its stability and practical utility for model development without retraining.

## 6 CONCLUSION

We introduced PhysVidBench, a benchmark for evaluating physical commonsense in T2V generation models. With a focus on tool use, object affordance, and causal interactions, the benchmark pairs physically enriched prompts with a structured evaluation pipeline that avoids the limitations of direct vision-language model scoring. Results across a range of open and closed-source models show that enriched prompts lead to meaningful gains, especially in larger systems, but also reveal persistent difficulties in spatial and temporal reasoning. These insights highlight the need for progress in architecture, training methods, and evaluation strategies that prioritize grounded physical understanding. PhysVidBench offers a foundation for tracking and accelerating such developments.

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

# A  APPENDIX

In this supplementary material, we provide:

- A comprehensive comparison of PhysVidBench against prior physical-commonsense video benchmarks, namely PhyGenBench, VBench 2.0, VideoPhy-2, and Impossible Videos, showing representative prompts, quantitative diversity statistics, and discussing key differences.

- A detailed description of our benchmark design and evaluation pipeline, covering prompt generation and upsampling with full LLM instruction templates.

- An expanded review of our physical commonsense ontology, including definitions for each of the seven dimensions, counts of scenarios per category, and example base vs. upsampled prompts.

- A structured taxonomy of physically grounded failure modes, introducing three complementary axes (Action Complexity, Scientific Principle, and Object Functionality) to reveal *why* certain prompts are difficult and how models break down when tasks require multi-step procedures, non-obvious physics, or functional recontextualization of objects.

- A step-by-step account of our dense captioning and question-answering workflow, including the dimension-targeted AuroraCap prompts, representative multi-perspective captions, and the full text of the prompt used to solicit binary judgments from the language model.

- Statistical significance analysis based on 10k bootstrap confidence intervals, including base vs. upsampled comparisons and difficulty-tier stratification across models.

- A full end-to-end example evaluation on PhysVidBench.

- Representative qualitative examples that contrast successful and failed video generations across different models and dimensions, accompanied by key frames and QA verdicts.

- Full details of our human evaluation study, including interface screenshots, and comparison between human and LLM judgments.

- An analysis of the robustness of our automatic evaluation pipeline, examining failure cases, sensitivity to captioning variations, and consistency across repeated runs.

- A brief discussion of the scope and the limitations of our proposed PhysVidBench benchmark.

# B EXPANDED COMPARISON WITH EXISTING BENCHMARKS

## B.1 REPRESENTATIVE PROMPTS FROM EACH BENCHMARK

While prior video-generation benchmarks probe particular aspects of physical reasoning, none offer the breadth and task-oriented focus of PhysVidBench.

- **PhyGenBench** (Meng et al., 2024) targets isolated physical laws like buoyancy, stress, elasticity through scenarios such as a sponge being squeezed or a ball bouncing. Its prompts highlight single phenomena in passive settings and omit any agent-driven tool use or multi-step manipulations.
- **VBench 2.0** (Zheng et al., 2025) introduces a wide variety of scene types and evaluates models on spatial consistency, temporal coherence, and visual realism. However, it devotes limited attention to object affordances or the mechanics of purposeful interactions, testing where objects "are" rather than what agents can "do" with them.
- **VideoPhy-2** (Bansal et al., 2025) emphasizes motion plausibility and physical violations, e.g. a Segway traversing speed bumps or a gymnast back-flipping near a wall. Yet, these scenarios rarely involve purposeful tool handling or causal sequences of actions.
- **Impossible Videos** (Bai et al., 2025b) leverages visually striking, intentionally implausible phenomena such as endless pouring of liquid, and self-moving books to expose hallucinations in generation models. While effective at uncovering obvious physics failures, its surreal setups lack grounding in real-world, goal-driven tasks.

In contrast, **PhysVidBench** is built around grounded, goal-oriented prompts derived from the PIQA dataset. Each scenario requires agents to use or repurpose everyday objects in deliberate sequences that simultaneously engage multiple commonsense dimensions (force application, spatial fit, material properties, containment, and procedural causality). This affordance-centric design makes PhysVidBench uniquely suited for assessing whether text-to-video models truly "understand" how tools and materials behave in practical, manipulable tasks.

In the following, we list representative prompts from each benchmark to illustrate these contrasts.

---

**PhyGenBench**

**Example Prompt 1:** A cup of water is slowly poured out in the space station, releasing the liquid into the surrounding area.

*Tests gravity absence in a synthetic environment; lacks real-world, goal-directed action or interaction with tools.*

- - - - - - - - - - - - - - - - - - - - - - - - - - - - - - - - - - - - - - - - - - - - - - - - - - -

**Example Prompt 2:** A timelapse of a water-filled sponge being forcefully squeezed by hand, with the pressure intensifying rapidly over time.

*Shows material deformation under stress, but the action is passive. It does not involve affordance-based manipulation or multi-step use of the sponge.*

- - - - - - - - - - - - - - - - - - - - - - - - - - - - - - - - - - - - - - - - - - - - - - - - - - -

**Example Prompt 3:** A vibrant, elastic rubber ball is thrown forcefully towards the ground, capturing its dynamic interaction with the surface upon impact.

*Focuses on elasticity and rebound, but lacks physical goal structure, repurposed tool logic, or causal task sequences.*

---

**VBench 2.0**

**Exmple Prompt 1:** A brown dog is on the left of an apple, then the dog moves to the right of the apple.

*Tests spatial consistency from different perspectives but lacks physical interaction, causality, or manipulation involving materials or tools.*

- - - - - - - - - - - - - - - - - - - - - - - - - - - - - - - - - - - - - - - - - - - - - - - - - - -

**Example Prompt 2:** A jar of peanut butter is opened in the space station, with the viscous liquid

---

slowly dispersing.

*Depicts fluid behavior in zero gravity but is passive and context-specific. There is no goal-directed manipulation or tool-based interaction.*

- - - - - - - - - - - - - - - - - - - - - - - - - - - - - - - - - - - - - - - - - - - - - - - - - - - -

**Example Prompt 3:** A person is drinking a glass of water, then they suddenly start cleaning the windows.

*Shows a behavioral transition but omits the physical process involved in cleaning. The objects used and the steps required are unspecified and ungrounded.*

---

**VideoPhy-2**

**Example Prompt 1:** A Segway bumps over a series of small speed bumps, the rider maintaining balance with slight adjustments.

*Evaluates motion plausibility and equilibrium but lacks goal-directed object use, sequential interaction steps, or any reasoning about tool affordances.*

- - - - - - - - - - - - - - - - - - - - - - - - - - - - - - - - - - - - - - - - - - - - - - - - - - - -

**Example Prompt 2:** A backflip is performed close to a wall. The person carefully ensures their momentum keeps them clear from hitting the wall.

*Involves physical trajectory reasoning but does not require interaction with any objects or manipulation of the environment for task completion.*

- - - - - - - - - - - - - - - - - - - - - - - - - - - - - - - - - - - - - - - - - - - - - - - - - - - -

**Example Prompt 3:** A backhoe digs a large hole in the ground, throwing up a mound of dirt.

*Depicts mechanical tool usage, yet the prompt remains high-level and underspecified. It does not describe the procedural steps or physical constraints involved in the interaction.*

---

**Impossible Videos**

**Example Prompt 1:** A person continuously pours water from a glass pot, yet remarkably the water level inside remains constant. This photo-realistic illusion defies natural physics as the endless stream flows without depleting the pot's contents. The scene takes place against a plain background, creating a focused view of this seemingly impossible phenomenon.

*Intentionally violates conservation of mass for visual paradox, but lacks grounded, goal-driven interaction or affordance-sensitive manipulation.*

- - - - - - - - - - - - - - - - - - - - - - - - - - - - - - - - - - - - - - - - - - - - - - - - - - - -

**Example Prompt 2:** A metallic iron ball bounces off a polished marble floor in a physically unusual manner. Defying normal physics, each consecutive bounce reaches a greater height than the last, creating an eerily unnatural progression of increasing rebounds. The glossy surface of the floor reflects the ball's motion in this photo-realistic scene, set against what appears to be an indoor space.

*Targets energy conservation violation in isolation. However, it contains no agent, object use, or causal physical interaction sequences grounded in real-world affordances.*

- - - - - - - - - - - - - - - - - - - - - - - - - - - - - - - - - - - - - - - - - - - - - - - - - - - -

**Example Prompt 3:** A book mysteriously opens by itself on a plain wooden desk surface, its pages flipping autonomously without any visible human intervention or external force. The photo-realistic footage captures this unexplainable movement in a well-lit indoor setting, defying natural physics as the book cover and pages lift and turn on their own.

*Demonstrates spontaneous motion but offers no agent–object interaction, mechanical force reasoning, or procedural physicality that would be necessary in task-driven video generation.*

---

**PhysVidBench (Ours)**

**Example Prompt 1:** A slice of bread is pressed onto small glass fragments scattered on a surface, picking them up.

*Involves soft material usage for delicate cleanup. Engages reasoning about texture, surface adhesion, and containment, grounded in everyday physical affordances.*

- - - - - - - - - - - - - - - - - - - - - - - - - - - - - - - - - - - - - - - - - - - - - - - - -

**Example Prompt 2:** Duct tape is wrapped around a stuck light bulb in a socket, and a hand uses the tape to twist the bulb counter-clockwise, loosening it.

*Demonstrates tool repurposing and torque application. Highlights grip mechanics, flexibility, and rotational force in a functional task context.*

- - - - - - - - - - - - - - - - - - - - - - - - - - - - - - - - - - - - - - - - - - - - - - - - -

**Example Prompt 3:** A drink can is placed upright into the opening of a shoe resting on the floor of a car's passenger side.

*Tests reasoning about shape compatibility, friction, and containment. Showcases improvised affordance in a constrained physical setup.*

---

## B.2 DATASET DIVERSITY AND SCALE

PhysVidBench achieves high conceptual coverage despite its compact size. Each prompt simultaneously probes several aspects of everyday physical reasoning such as affordances, causality, material response, and procedural manipulation, providing richer supervision density than benchmarks focused on isolated physical laws. As shown in Table 6, both the base and upsampled prompts exhibit substantially stronger linguistic and physical diversity (unique verbs, nouns, and action–object pairs) than prior benchmarks, reflecting a broad range of real-world scenarios.

Table 6: **Linguistic and physical diversity comparison across benchmarks.** PhysVidBench (base and upsampled variants) shows substantially higher diversity in verbs, nouns, and action–object pairs than prior physics-oriented video benchmarks, reflecting broader coverage of real-world physical interactions despite its compact prompt set.

| Benchmark | # Prompt | Unique Verbs | Verbs per Prompt | Unique Nouns | Nouns per Prompt | Action–Object Pairs per Prompt | Lexical Density |
|---|---|---|---|---|---|---|---|
| PhyGenBench | 160 | 77 | 0.481 | 239 | 1.494 | 0.79 | 0.55 |
| PhyGenBench (explicit prompt) | 160 | 232 | 1.45 | 408 | 2.55 | 2.91 | 0.57 |
| VideoPhy-2 (train set) | 3343 | 773 | 0.231 | 2111 | 0.631 | 1.43 | 0.59 |
| VBench 2.0 | 1013 | 593 | 0.585 | 1282 | 1.266 | 1.75 | 0.55 |
| VBench 2.0 (augmented) | 1013 | 1195 | 1.18 | 2979 | 2.941 | 6.37 | 0.61 |
| PhysVidBench (base) | 383 | 262 | 0.684 | 708 | 1.849 | 2.03 | 0.60 |
| PhysVidBench (upsampled) | 383 | 623 | 1.627 | 1233 | 3.219 | 6.68 | 0.64 |

Although the benchmark contains 383 core prompts, the effective evaluation scale is large: across models, captions, and physics-grounded questions, PhysVidBench generates more than 42,000 QA instances per split, offering fine-grained diagnostic resolution in practice.

Finally, the benchmark's scale aligns closely with established physics-oriented diagnostic datasets (PhyGenBench: 160 prompts; VideoPhy: 688 prompts), which similarly prioritize conceptual precision and interpretability over raw quantity. PhysVidBench follows this design philosophy to ensure that the benchmark remains both computationally feasible for the community and rich enough to meaningfully evaluate physical commonsense in T2V models.

# C  BENCHMARK DESIGN AND EVALUATION DETAILS

## C.1  PROMPT GENERATION AND UPSAMPLING

Our benchmark construction pipeline begins with the PIQA dataset, utilizing its extensive repository of physically grounded scenarios. Specifically, we use the training and validation splits and exclude the test split due to the absence of ground truth labels. By selecting only correct solutions from these splits, we obtain more than 17,000 goal-solution pairs. Our pipeline then proceeds through three stages: (1) *Initial filtering*; (2) *Video prompt generation*; and (3) *Prompt upsampling*.

**Stage 1: PIQA Dataset and Initial Filtering.** We use Gemini 2.5 Pro model to select only those PIQA scenarios that involve secondary or repurposed tool use, those that inherently require physical-commonsense reasoning. This filtering yields a compact set of complex, interaction-centric tasks ideal for text-to-video evaluation. Below is an example prompt illustrating the instructions provided to Gemini 2.5 Pro in this stage.

---

**Video Generation Pipeline - Stage 1: PIQA Dataset and Initial Filtering**

I will provide a series of examples from the PIQA dataset where common household objects or tools are used either in their primary/intended way or in repurposed/alternative ways (i.e., "life hacks"). Your task is to determine whether a given PIQA goal–solution pair reflects the primary usage of an object or not.

**Definition**: If the object is being used in its intended, standard, or conventional way, respond with: No
If the object is being used in an alternative or repurposed way (i.e., not its typical function), respond with: Yes

You are given a list of goal–action pairs (e.g., "Water Bottle → Egg Separator", "Credit Card → Ice Scraper") which illustrate how PIQA questions often involve creative or secondary uses of common items. These examples represent secondary usage. This list can guide your understanding of what constitutes repurposing. You will now be given a PIQA goal and two solutions. Based on your understanding of object affordances and typical usage. Determine only whether the goal/solution pair represents a secondary use of an object. Write only the answer, Yes or No. Do not explain or describe the reasoning.

Here are examples from the PIQA dataset where common objects or tools are used for a purpose other than their primary, intended function (i.e., alternative uses or "life hacks"):

**Water Bottle → Egg Separator:**
**Goal:** To separate egg whites from the yolk using a water bottle...
**Action:** Squeeze the water bottle and press it against the yolk. Release, which creates suction and lifts the yolk.

**Hair Net → Vacuum Filter:**
**Goal:** How do I find something I lost on the carpet?
**Action:** Put a hair net on the end of your vacuum and turn it on. (To catch small items before they go into the vacuum bag).

These examples demonstrate the kind of creative repurposing and understanding of object affordances beyond their primary function that the PIQA dataset aims to capture.
Write only the answer for each goal with comma seperator: Yes or No. Do not explain or describe the reasoning.

Give the answer for this Goal Solution pair:
**Goal:** How can I paint stars on a black background?
**Solution:** Use an old toothbrush covered in white paint and flick at the background.

---

**Stage 2 - Video Prompt Generation.** We use the Gemini 2.5 Pro model to transform each filtered goal–solution pair into a concise, visually explicit video generation prompt. This process produces 383 distinct *base* prompts, each outlining a self-contained scenario with clear, physically grounded actions. Below is an example prompt illustrating the instructions provided to Gemini 2.5 Pro in this stage.

---

**Video Generation Pipeline - Stage 2: Video Prompt Generation**

You are given question–answer pairs from the PIQA dataset. Each pair consists of a physical goal (what someone wants to achieve) and a solution (how to achieve it). Your task is to write a **realistic and physically plausible video generation prompt** that visually demonstrates the provided solution.

**Important Instructions:**

- Only generate a short, visualizable **video prompt** that clearly shows the solution being carried out.

- The prompt should describe **a realistic short video clip** (under 10 seconds).

- You do **not** have to begin each prompt with "A person". Be natural — use whatever fits best (e.g., "A dog. . . ", "Hands. . . ", "Someone. . . ", "Two people. . . ", "The object. . . ").

- If the solution is not visualizable or does not reflect meaningful physical interaction or common-sense action, **skip it** (do not generate a prompt).

- Be **specific** and **physically grounded** — actions should be directly observable (e.g., cutting, pouring, lifting, placing).

- Do not repeat the goal or the original solution text.

**Example Conversions:**

**Goal:** You want to dry your wet hands.
**Solution:** Use a towel
**Video Prompt for Solution:** Wet hands are rubbed against a cotton towel.

**Goal:** You want to keep a door open.
**Solution:** Use a doorstop
**Video Prompt for Solution:** A door is held open by sliding a rubber doorstop underneath.

**Goal:** You want to get rid of wrinkles in a shirt.
**Solution:** Use an iron
**Video Prompt for Solution:** An iron glides over a wrinkled shirt on an ironing board, flattening it.

Now, for each of the following goal–solution pairs from the PIQA dataset, return a single line in the format below **only if it qualifies**:

PIQA Input:
**Goal:** How can I paint stars on a black background?
**Solution:** Use an old toothbrush covered in white paint and flick at the background.

---

**Stage 3 - Prompt Upsampling.** We use the Gemini 2.5 Pro model to enrich each *base* prompt with fine-grained physical details such as explicit force application, material behaviors, and spatial constraints, while preserving the original scene scope. This controlled upsampling yields 383 *upsampled* prompts that surface deeper commonsense cues without introducing new objects or outcomes. Below is an example prompt illustrating the instructions provided to Gemini 2.5 Pro in this stage.

---

**Video Generation Pipeline - Stage 3: Prompt Upsampling**

You are a reasoning-augmented AI model tasked with upsampling video generation prompts using knowledge of physical commonsense and object affordances, while strictly preserving the original scene boundaries. Given a simple prompt, you must rewrite it into a richer, more physically grounded version by:

- Elaborating on object properties (e.g., material, size, rigidity, weight) that affect how actions are performed
- Clarifying affordances (e.g., what can be held, moved, poured, tied) based on those properties
- Expanding the action description in a physically realistic, causally plausible sequence
- Including any relevant environmental or contextual conditions (e.g., gravity, surface type)
- Do not introduce any new objects or tools not mentioned in the original prompt
- Do not describe outcomes or events that happen after the main action (no result, consequence, or future speculation)
- Your goal is to make the prompt maximally useful for video generation models that rely on physical reasoning, without hallucinating or extending beyond the original scope.

**Example Input:**
"A person places coffee filters inside the cup holders of a car."

**Upsampled Output:**
"A person carefully pushes white paper coffee filters, each lightweight and slightly crinkled, into the round plastic cup holders on the car's center console. The filters conform to the shape of the holders due to their flexible, foldable structure. The car interior is clean and the cup holders are empty, allowing the filters to settle flush against the bottom surface."

Only return the upsampled prompt as output. Do not explain your reasoning or include anything else in your answer.

**Prompt:** A person dips an old toothbrush in white paint and flicks it onto a black background, creating stars.

---

Altogether, PhysVidBench comprises 766 unique prompts (383 base + 383 upsampled), enabling a comprehensive evaluation of physical commonsense reasoning in text-to-video models. Representative examples illustrating the outcome of each stage of our pipeline are shown below.

---

**Example-1**

**Stage 1: PIQA Dataset and Initial Filtering**
Goal: How can I paint stars on a black background?
Solution: Use an old toothbrush covered in white paint and flick at the background.

- - - - - - - - - - - - - - - - - - - - - - - - - - - - - - - - - - - - - - - - - - - - - - - - - - - - - - - - - - - - - - - -

**Stage 2: Video Prompt Generation**
A person dips an old toothbrush in white paint and flicks it onto a black background, creating stars.

- - - - - - - - - - - - - - - - - - - - - - - - - - - - - - - - - - - - - - - - - - - - - - - - - - - - - - - - - - - - - - - -

**Stage 3: Prompt Upsampling**
A person firmly grips the handle of an old, worn toothbrush, its plastic slightly yellowed with age. They dip the frayed bristles into a can of thick, white paint. The paint clings to the bristles due

---

to its viscosity. With a controlled wrist motion, they flick the paint-laden toothbrush towards a large, matte black canvas backdrop. Tiny droplets of white paint detach from the bristles and spray outwards, propelled by the force of the flick. The droplets travel through the air, influenced by gravity, before adhering to the vertical surface of the canvas.

---

**Example-2**

**Stage 1: PIQA Dataset and Initial Filtering**
Goal: When I am driving an older car with no cupholders where can I place my beverage?
Solution: Use an old shoe, place the drink container in the place where your foot enters the shoe and it will help stabilize it.

------------------------------------------------------------

**Stage 2: Video Prompt Generation**
A drink can is placed upright into the opening of a shoe resting on the floor of a car's passenger side.

------------------------------------------------------------

**Stage 3: Prompt Upsampling**
A cylindrical aluminum drink can, rigid due to its metal construction and filled with liquid giving it noticeable weight, is carefully maneuvered vertically downwards. It is inserted, base-first, into the flexible fabric opening of a casual shoe which is resting flat on its firm sole upon the textured floor surface of the car's passenger side. The can maintains its upright orientation under gravity as its smooth, flat circular base makes stable contact with the interior insole surface deep within the shoe's cavity.

---

**Example-3**

**Stage 1: PIQA Dataset and Initial Filtering**
Goal: How do I find something I lost on the carpet?
Solution: Put a hair net on the end of your vacuum and turn it on.

------------------------------------------------------------

**Stage 2: Video Prompt Generation**
A person attaches a hair net to the end of a vacuum cleaner and uses it to search for a small object on a carpet.

------------------------------------------------------------

**Stage 3: Prompt Upsampling**
A person stretches the elastic rim of a fine, white nylon hair net, expanding its opening, and carefully fits it over the circular nozzle of a dark gray plastic vacuum cleaner hose. The net conforms to the hose's shape, held in place by its own tension. The vacuum cleaner is then used to gently hover over a low-pile carpet, its fibers slightly compressed by the vacuum's suction. The person guides the nozzle in slow, overlapping passes, the hair net acting as a barrier to prevent small objects from being sucked into the machine.

## C.2   Physical Commonsense Dimensions and Sample Prompts

Below we present each of the seven physical commonsense dimensions defined in our benchmark, accompanied by representative base and upsampled prompts from PhysVidBench. While the prompts shown under each category were selected because they most strongly emphasize the associated reasoning dimension, it is important to note that prompts in PhysVidBench are not confined to a single category. In practice, most scenarios engage multiple forms of physical reasoning at the same time. For example, a prompt may involve both force application and spatial configuration, or material properties combined with temporal sequencing. This multi-dimensional grounding reflects the complex and realistic nature of physical tasks, and it distinguishes PhysVidBench from benchmarks that test physical concepts in isolation.

**Fundamental Physics:** *The study of inherent object attributes, including material composition, rigidity, softness, and functional affordances like containment or support*

**Base Prompt:** A flashlight lying on a dark floor illuminates tiny shards of broken glass by casting long shadows from them.

**Upsampled Prompt:** A cylindrical metal flashlight rests horizontally on a dark, flat floor, its switch in the 'on' position. A focused beam of bright light emits from its lens, traveling parallel and close to the floor surface. This low-angle light strikes multiple tiny, sharp-edged, transparent shards of broken glass scattered across the floor. Because the light source is nearly level with the floor, even the small vertical profile of each glass shard is sufficient to block the light path, causing disproportionately long, thin shadows to stretch out behind each shard, starkly visible against the surrounding darkness of the floor.

- - - - - - - - - - - - - - - - - - - - - - - - - - - - - - - - - - - - - - - - - - - - - - - - - - -

**Base Prompt:** A hand pours a small amount of white flour into a larger bowl of water and begins stirring with a spoon.

**Upsampled Prompt:** A human hand directs a small stream of fine, white, lightweight flour powder downwards due to gravity, letting it fall onto the surface of still water held within a larger, rigid bowl resting on a flat surface. The flour particles make contact with the liquid. The hand then grips the rigid handle of a spoon, submerges the spoon's end into the water where the flour landed, and initiates a stirring motion, moving the spoon through the water and the suspended flour particles.

- - - - - - - - - - - - - - - - - - - - - - - - - - - - - - - - - - - - - - - - - - - - - - - - - - -

**Base Prompt:** A plastic bag containing meat sits on an upside-down aluminum pot, with another pot filled with water placed directly on top of the bagged meat.

**Upsampled Prompt:** A flexible, translucent plastic bag, visibly holding a weighty piece of raw meat which causes the thin material to bulge slightly, is positioned on the flat, circular, upturned bottom of a lightweight yet rigid aluminum cooking pot resting stably upside-down on its rim. Placed directly centered on top of this bagged meat rests another rigid cooking pot, its flat bottom making full contact; this second pot contains a significant volume of water, adding substantial downward weight due to gravity. The pliable plastic bag is compressed between the base of the top, water-filled pot and the upturned base of the bottom pot, conforming slightly to the shapes above and below it while transferring the load.

**Object Properties & Affordances:** *The study of inherent object attributes, including material composition, rigidity, softness, and functional affordances like containment or support*

**Base Prompt:** A stainless steel bucket is placed upside down over a small ant crawling on the ground.

**Upsampled Prompt:** Human hands carefully grasp several thin, lightweight, and brittle potato chips, noting their fragile structure. The hands move towards the inside of a metal barbecue grill where a pile of dark grey, blocky charcoal briquettes rests. The briquettes have a rough, porous texture and form an uneven surface. The hands gently lower and release the delicate potato chips directly onto the top surfaces of the stacked briquettes, allowing gravity to settle the light chips precariously onto the coarse, irregular shapes without applying significant pressure that would cause them to crumble.

- - - - - - - - - - - - - - - - - - - - - - - - - - - - - - - - - - - - - - - - - - - - - - - - - - -

**Base Prompt:** A hand rubs a piece of bread back and forth over pencil writing on paper.

**Upsampled Prompt:** A human hand grips a piece of soft, somewhat pliable white bread, likely without crusts, between its fingers. Positioned over a sheet of standard writing paper lying flat on a hard surface, the hand methodically rubs the bread's yielding, slightly textured underside back and forth. This motion occurs directly over an area marked with fine, gray pencil writing, causing the bread's surface to make repeated contact with the graphite lines on the paper under consistent, light pressure.

- - - - - - - - - - - - - - - - - - - - - - - - - - - - - - - - - - - - - - - - - - - - - - - - - - -

**Base Prompt:** Hands fold one short end of a credit card up and the other short end down, then place a smartphone onto the upward fold.

**Upsampled Prompt:** A pair of hands holds a standard-sized, thin, rectangular plastic credit

card. Applying pressure with fingers, the hands bend one short edge of the rigid-yet-flexible card sharply upwards along its width, creating a defined crease. Then, the hands bend the opposite short edge sharply downwards along its width, forming a second parallel crease, configuring the card into a small S-shaped structure. Next, the hands carefully lift a smooth, relatively heavy, rigid rectangular smartphone. Aligning the bottom edge of the phone with the small, upward-facing plastic ledge created by the first fold on the credit card, the hands gently lower the weight of the smartphone onto this narrow support.

**Spatial Reasoning:** *The ability to interpret and infer spatial relations, including position, geometry, occlusion, orientation, and fit between objects in a scene*

**Base Prompt:** The tip of a spoon presses a square outline into a slice of bread, then the back of the spoon pushes down the center.
**Upsampled Prompt:** The hard, pointed metal tip of a rigid spoon presses sequentially into the soft, porous surface of a slice of bread, tracing a square outline by creating shallow indentations. The bread yields slightly under the focused pressure of the tip. Subsequently, the smooth, curved underside (back) of the spoon's bowl is centered within the traced square. Applying downward force, the broader surface of the spoon back pushes into the bread, compressing the central area significantly more deeply than the initial outline, creating a distinct depression within the square boundary on the yielding bread slice.

- - - - - - - - - - - - - - - - - - - - - - - - - - - - - - - - - - - - - - - - - - - - - - - - - - -

**Base Prompt:** Crushed egg shells and water are shaken vigorously inside a clear bottle.
**Upsampled Prompt:** Inside a clear, rigid bottle containing clear liquid water, numerous small, lightweight, brittle fragments of crushed egg shells are forcefully agitated. The bottle undergoes vigorous, rapid shaking motions, causing the water to slosh violently and carry the sharp-edged shell fragments in turbulent, swirling patterns within the confined space. The mixture of water and shell fragments repeatedly impacts the inner surfaces of the bottle due to the applied force.

- - - - - - - - - - - - - - - - - - - - - - - - - - - - - - - - - - - - - - - - - - - - - - - - - - -

**Base Prompt:** A hand pushes a plastic drinking straw upwards through the bottom of a strawberry, popping the green stem out the top.
**Upsampled Prompt:** A human hand firmly grips the lower end of a thin, cylindrical, rigid plastic drinking straw. The straw's end is pressed against the pointed bottom tip of a ripe, red strawberry, which is soft enough to be pierced but firm enough to maintain its shape. The hand applies a continuous, steady upward force along the axis of the straw. Due to its rigidity and the applied pressure, the straw penetrates the yielding flesh of the strawberry, moving vertically upwards through its center. As the leading end of the straw travels inside the fruit towards the wider top where the green, leafy stem is attached, it pushes against the stem's base from below. This focused upward force exerted by the straw dislodges the relatively light stem structure, causing it to pop out from the top surface of the strawberry.

**Temporal Dynamics:** *The domain addressing timing, sequencing, and the causal structure of events over time, such as ordering, delays, and waiting*

**Base Prompt:** A plastic bucket is placed upside down on someone's head like a hat.
**Upsampled Prompt:** A lightweight, rigid plastic bucket, oriented upside down so its solid bottom faces upwards, is carefully lowered onto a person's head. The open circular rim of the bucket makes contact and rests stably upon the crown of the head, balanced there by gravity. The bucket's inherent rigidity allows it to maintain its shape, covering the top of the head like a large, hollow hat.

- - - - - - - - - - - - - - - - - - - - - - - - - - - - - - - - - - - - - - - - - - - - - - - - - - -

**Base Prompt:** A child places their hand on a large circle sticker affixed to the side of a car in a parking lot.
**Upsampled Prompt:** A small child extends their arm and carefully places their soft, open hand flat onto the smooth surface of a large, circular sticker. The thin, flexible sticker is firmly adhered by its backing to the hard, vertical metal side panel of a stationary car. The car is situated on the level asphalt ground of an outdoor parking lot. The child's palm and fingers press lightly against

the sticker's surface, conforming slightly to the car's panel curvature beneath it.

- - - - - - - - - - - - - - - - - - - - - - - - - - - - - - - - - - - - - - - - - - - - - -

**Base Prompt:** A hand squeezes toothpaste onto a cloth and rubs it in circles over a hazy car headlight, leaving a thin white coating.

**Upsampled Prompt:** A human hand squeezes a dollop of viscous, opaque white toothpaste directly onto a section of a soft, flexible fabric cloth. The paste adheres to the cloth's surface. The hand then presses this part of the cloth firmly against the hard, smooth, but visibly hazy surface of a car's plastic headlight cover. Maintaining steady pressure, the hand rubs the cloth in continuous circular motions over the hazy area. The pliable cloth conforms to the headlight's curvature, spreading the toothpaste evenly as it's rubbed, leaving behind a thin, uniform white coating on the headlight surface.

---

**Action & Procedural Understanding:** *The comprehension of structured, goal-driven sequences of actions involving procedural steps toward task completion*

**Base Prompt:** Hands press a piece of clay over the joined edges of pieces on top of a bowl.

**Upsampled Prompt:** Human hands hold a separate lump of soft, malleable, slightly damp clay. The hands carefully position this clay lump directly over the visible seam where the edges of several harder, pre-joined clay pieces meet. These pieces rest securely on the upward-facing surface of a rigid ceramic bowl. Fingers then apply firm, steady downward pressure onto the soft clay lump, causing it to flatten and spread smoothly across the underlying seam, adhering to the surfaces of the joined pieces due to its pliable and slightly sticky nature.

- - - - - - - - - - - - - - - - - - - - - - - - - - - - - - - - - - - - - - - - - - - - - -

**Base Prompt:** A cloth damp with olive oil is wiped over a metal key. The key is then inserted smoothly into a keyhole.

**Upsampled Prompt:** A soft, absorbent cloth, damp and glistening slightly with viscous olive oil, is wiped firmly across the surfaces of a rigid, metallic key. The pliable cloth presses into the key's teeth and grooves, coating the hard metal with a thin, lubricating film of oil. Immediately after, the oil-coated metal key is precisely aligned with the opening of a metal keyhole and inserted with a steady, linear motion, sliding effortlessly into the lock mechanism due to the oil reducing friction between the contacting metal surfaces.

- - - - - - - - - - - - - - - - - - - - - - - - - - - - - - - - - - - - - - - - - - - - - -

**Base Prompt:** Hands carefully pour a green liquid over the back of a spoon into a shot glass containing a brown liquid, forming a distinct layer. Then, a cream liquid is poured similarly on top.

**Upsampled Prompt:** Hands carefully position a rigid metal spoon, its convex back facing upwards, just above the surface of a dark brown, relatively dense liquid resting at the bottom of a small, clear glass shot glass. With precise control, a vibrant green liquid, less dense than the brown liquid, is steadily poured onto the spoon's curved back. The spoon's smooth surface guides the green liquid's flow, allowing it to gently cascade onto the brown liquid's surface, minimizing turbulence and forming a distinct, level layer without significant mixing. Following this, the hands adjust the spoon's position slightly upwards, holding it just over the newly formed green layer. Then, an opaque, cream-colored liquid, possessing an even lower density than the green liquid, is poured using the same technique over the back of the spoon, flowing down gently to create a third, clearly separated layer resting atop the green one within the confines of the shot glass's rigid walls.

---

**Material Interaction & Transformation:** *The study of how materials respond to external forces or processes, including transformations like melting, freezing, breaking, or chemical change*

**Base Prompt:** Hands hold a smartphone inside a clear ziploc bag, tapping the screen while raindrops fall on the bag.

**Upsampled Prompt:** Human hands firmly grip the sides of a clear, flexible plastic ziploc bag, holding it steady. Inside the sealed bag rests a rigid, rectangular smartphone with its glass screen visible. A finger repeatedly taps onto the thin, pliable plastic surface directly overlying the phone's touch-sensitive screen, the pressure transmitting through the material. Simultaneously,

small water droplets, falling under gravity, land and bead up on the exterior surface of the waterproof bag.

------------------------------------------------

**Base Prompt:** Water flows from a sink faucet into the scoop of a dustpan, travels down its hollow handle, and pours into a bucket sitting beside the sink.

**Upsampled Prompt:** Clear water flows downwards in a steady stream from the metal nozzle of a sink faucet, driven by gravity. The stream lands directly onto the rigid, concave surface of a plastic dustpan's scoop, which is positioned underneath the faucet. Due to the angle of the dustpan, the accumulating water is channeled towards the opening at the base of the scoop leading into its hollow handle. The water then travels downwards through the enclosed tubular space within the handle. The open end of the handle is positioned directly over the wide opening of a sturdy plastic bucket resting stationary on the floor beside the sink, allowing the water exiting the handle to fall vertically into the bucket.

------------------------------------------------

**Base Prompt:** A golf ball is placed on a countertop; it rolls slightly, then the counter is subtly adjusted, and the ball comes to a stop.

**Upsampled Prompt:** A small, dense, hard golf ball with a characteristic dimpled texture is gently placed onto a large, flat, smooth countertop surface. Because of its perfectly spherical shape and the countertop potentially being minutely uneven, the heavy ball starts to roll slowly across the rigid surface. Subsequently, the countertop experiences a subtle, barely perceptible shift in its orientation, adjusting its levelness. This slight change counteracts the initial impetus for rolling, altering the gravitational pull along the surface, and the ball, influenced by friction and the new level plane, decelerates smoothly until it comes to a complete standstill upon the countertop.

**Force and Motion:** *The domain focused on physical interactions governed by forces, including motion, pushing, pulling, lifting, and properties like inertia*

**Base Prompt:** A strong magnet is slowly moved across a drywall surface, suddenly snapping and sticking to the wall over a stud location.

**Upsampled Prompt:** A dense, rigid metallic magnet, possessing a strong magnetic field, is held close to a vertical, painted drywall surface. It is guided slowly and steadily in a continuous motion across the slightly textured face of the wall. As the path of the strong magnet crosses over the position of a hidden ferromagnetic metal stud located just behind the thin drywall layer, the attractive magnetic force increases sharply and suddenly. This abruptly intensified pull overcomes the controlled movement, causing the heavy magnet to accelerate rapidly towards the wall, making firm contact and sticking securely against the drywall surface directly above the unseen stud's location due to the powerful magnetic attraction.

------------------------------------------------

**Base Prompt:** Soft tortillas are draped and wedged between the bumps of an upside-down muffin pan that has been sprayed with oil.

**Upsampled Prompt:** Soft, pliable, circular tortillas, thin and flexible due to their material properties, are draped over the rigid structure formed by an upside-down metal muffin pan. Gravity pulls the lightweight tortillas downwards. They are then carefully pressed and wedged into the concave spaces between the protruding, rounded bumps of the pan. The tortillas bend and conform to the curved shape of these spaces. The pan's surface, both bumps and crevices, is coated in a visible layer of slick cooking oil, facilitating the placement and slight sliding of the tortillas as they are settled into position.

------------------------------------------------

**Base Prompt:** A hand places a colorful silicone cupcake liner into the empty cup holder of a car's center console.

**Upsampled Prompt:** A hand gently pinches a colorful, flexible silicone cupcake liner between thumb and forefinger. The liner, lightweight and pliable with distinct fluted sides, is carefully lowered vertically into an empty, cylindrical cup holder recessed into the car's rigid plastic center console. The liner slides smoothly against the inner wall and comes to rest flat against the bottom surface of the holder, its flexible structure allowing it to fit snugly within the defined space.

## C.3 EVALUATION DETAILS

To evaluate physical commonsense on PhysVidBench, we employ a three-stage pipeline: (1) generate targeted yes/no questions from each upsampled prompt, (2) produce multi-perspective dense captions for the corresponding video, and (3) use an LLM to answer those questions based solely on the captions.

**Stage 1: Physics-Grounded Question Generation.** For each upsampled prompt, we prompt Gemini 2.5 Pro to craft precise yes/no questions that probe our seven physical-commonsense dimensions (e.g., "Did the person apply sufficient force to separate the objects?"). Every question is guaranteed to have a "Yes" answer and is answerable by inspecting the prompt alone. We fix and store this question set for downstream evaluation of each model's video output. Below is an example prompt used in this stage.

---

**Sample Prompt for Evaluation - Stage 1: Physics-Grounded Question Generation**

Generate as many diverse, concrete yes/no questions as possible based only on the provided prompt. Each question must:
- Be specific and answerable solely from the prompt (no outside assumptions)
- Focus on physical or procedural understanding, not abstract or emotional reasoning
- Be non-redundant — do not repeat similar ideas or reword the same question
- Cover a wide range of physical reasoning categories
- Include a balanced mix of Yes and No answers
- If a question relates to multiple reasoning categories, include all applicable types

**Physical Reasoning Categories:** (Choose these types, write only the category like Fundamental Physics, Object Properties etc. )
- **Fundamental Physics:** energy, causality, equilibrium, state change
- **Object Properties & Affordances:** material type, rigidity, softness, containment
- **Spatial Reasoning:** fit, position, occlusion, geometry, orientation
- **Temporal Dynamics:** ordering, timing, waiting, delays
- **Action & Procedural Understanding:** goal-directed behavior, methodical steps
- **Material Interaction & Transformation:** melting, freezing, breaking, chemical change
- **Force and Motion:** pushing, pulling, lifting, inertia

**Output Format:**
- Questions and Answers:
- Q: <yes/no question>?
- A: Yes / No
- Type: <Reasoning Category 1, Reasoning Category 2, ...>

**You must return:**
- A rich, diverse set of yes/no questions
- Each with a correct answer
- Each labeled with all relevant reasoning categories, separated by commas

Start with the following prompt:
**Prompt:** Person firmly grips the handle of an old, worn toothbrush, its plastic slightly yellowed with age. They dip the frayed bristles into a can of thick, white paint. The paint clings to the bristles due to its viscosity. With a controlled wrist motion, they flick the paint-laden toothbrush towards a large, matte black canvas backdrop. Tiny droplets of white paint detach from the bristles and spray outwards, propelled by the force of the flick. The droplets travel through the air, influenced by gravity, before adhering to the vertical surface of the canvas.

---

**Stage 2: Dense Video Captioning.** For each generated video, we use the AuroraCap model (Chai et al., 2025) to generate eight distinct captions: one general-purpose caption and seven dimension-specific captions (one per commonsense dimension). For instance, to highlight spatial relationships, we use the prompt "Describe the spatial layout and object positions. . . "; to surface force interactions we use "Describe how forces are applied and resisted. . . ." Constraining each caption to direct observations with a specific prompt at the end prevents hallucination and ensures complementary, multi-perspective descriptions. Below is the full template we provide to AuroraCap in this stage.

---

**Prompt for Evaluation - Stage 2: Dense Video Captioning**

- Describe the video in detail.

- Describe the physical principles at work in the video, such as energy transfer, causal relationships, balance or imbalance, and any visible changes in physical state and etc. (e.g., solid to liquid).

- Describe the main objects in the video focusing on their material properties (e.g., rigid, soft, metallic), and what actions they allow or prevent and etc. (e.g., can be squeezed, can contain something).

- Describe the spatial layout and relationships in the video: object positions, orientations, fit between shapes, occlusions, and how geometry affects interactions and etc..

- Describe the sequence and timing of events in the video, including any delays, waiting periods, or causal orderings between actions or state changes and etc..

- Describe the actions performed in the video, focusing on the goals of the agent, the order of steps taken, and whether the actions appear intentional or methodical and etc..

- Describe how materials change or interact in the video: melting, freezing, breaking, mixing, or undergoing chemical or physical transformations and etc..

- Describe how forces are applied in the video (e.g., pushing, pulling, lifting), how objects respond (e.g., acceleration, resistance), and any indications of inertia or physical resistance and etc..

Only describe what can be directly observed in the video. Do not make assumptions or include external knowledge that is not visually confirmed.

---

**Stage 3: LLM as a Judge.** We assemble a single text prompt that includes: (a) all yes/no questions generated in Stage 1, and (b) the eight dense captions produced in Stage 2. We pass this prompt to Gemini-2.5-Flash-Preview-04-17 (Google DeepMind, 2025b), instructing it to respond only with "Yes" or "No" for each question based solely on the provided captions. Below, we present the exact Stage 3 prompt used to evaluate the sample video generated by the Wan2.1 (14B) model from the Stage 1 prompt. This prompt combines the eight dense captions produced in Stage 2 with the corresponding physical-commonsense questions from Stage 1, illustrating how the LLM integrates multi-perspective textual evidence and targeted queries during evaluation.

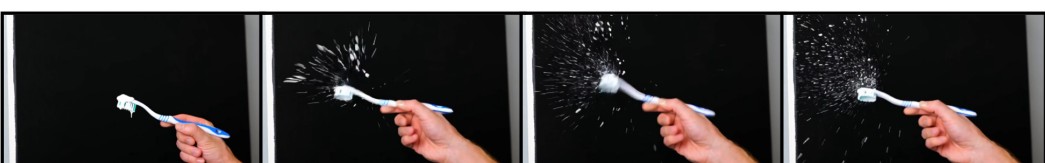

---

**Example Prompt for Evaluation - Stage 3: LLM as a Judge**

You are given 8 captions describing different aspects of the same video. Answer "Yes" ONLY if at least one caption supports it, otherwise answer "No".

**Caption 1:** The video captures a moment where a person is using a toothbrush to apply a white substance, likely toothpaste, onto a dark surface. The toothbrush is held in a way that suggests the person is in the process of brushing, with the bristles making contact with the

---

surface. The substance is being applied in a way that creates a textured effect, with some areas appearing more saturated than others. The background is a solid, dark color, providing a stark contrast to the bright white of the toothpaste. The lighting is even, allowing for clear visibility of the toothbrush and the substance being applied. The focus is on the action of brushing, with the background remaining unchanged, emphasizing the simplicity of the scene.

**Caption 2:** The video captures a moment where a toothbrush is being used to apply a white substance, likely toothpaste, onto a dark surface. The energy transfer is evident as the bristles of the toothbrush create a splatter effect, with the white toothpaste dispersing in a way that suggests a mix of solid and liquid components. The balance of the scene is slightly tilted, with the toothbrush being the focal point, while the dark surface provides a contrast that highlights the action. The physical state of the toothbrush and toothpaste is clearly observable; the toothbrush is in motion, and the toothpaste is in a semi-liquid state, transitioning from a solid to a liquid form as it spreads.

**Caption 3:** The main objects in the video are a toothbrush and a paintbrush. The toothbrush, with its soft bristles, is designed for cleaning teeth, allowing it to gently scrub surfaces. The paintbrush, with its metallic handle, is used for applying paint, preventing the paint from dripping and allowing for controlled application. The actions they allow are brushing and painting, respectively, while the toothbrush's soft bristles prevent abrasion and the paintbrush's handle allows for a firm grip.

**Caption 4:** The video captures a moment where a hand is applying a white substance, likely paint, to a dark surface using a toothbrush. The toothbrush is positioned diagonally across the frame, with the bristles facing the camera. The paint is being applied in a way that creates a textured effect, with some areas showing a smooth application and others showing a more uneven, brush-like texture. The hand is in the process of painting, with the fingers gripping the toothbrush and the thumb supporting it. The background is a solid, dark color, providing a stark contrast to the white paint, which enhances the visibility of the brush strokes. The overall layout is simple, focusing on the action of painting without any distractions.

**Caption 5:** The video captures a moment where a person is applying a white substance, likely paint, to a dark surface using a toothbrush. The action is deliberate and focused, with the bristles of the toothbrush making contact with the surface, creating a small, white splatter. The hand holding the toothbrush is steady, with fingers gripping the handle firmly. The background is a solid, dark color, providing a stark contrast to the white paint, which enhances the visibility of the splatter. The lighting is bright, casting a soft shadow of the toothbrush on the surface. The overall scene is static, with no movement or change in the environment, allowing the viewer to concentrate on the application process.

**Caption 6:** The video captures a moment where a person is applying a white substance, likely paint, to a dark surface using a toothbrush. The actions are deliberate and show a methodical approach, with the substance being carefully squeezed from a tube and then applied in a controlled manner. The agent's hand is steady, with fingers positioned to hold the tube and the toothbrush, indicating a focus on precision. The substance is being applied in a way that suggests a deliberate effort to create a specific effect, possibly for artistic or decorative purposes. The overall scene is calm and focused, with the agent's hand being the main point of interest.

**Caption 7:** The video captures a moment where a toothbrush is being used to apply a white substance, likely a cleaning agent, to a dark surface. The substance appears to be in a liquid state, as evidenced by the way it drips and pools on the surface. The toothbrush is held in a way that suggests it is being used to spread the substance, with the bristles making contact with the surface. The fingers of the hand holding the toothbrush are visible, with one finger slightly raised, indicating the action of applying pressure. The background is a solid, dark color, providing a stark contrast to the white substance, which emphasizes the cleaning process.

**Caption 8:** The video captures a moment where a toothbrush is being used to apply a white substance, likely toothpaste, to a dark surface. The bristles of the toothbrush are in contact with the surface, and the substance is being squeezed out, creating a small puddle. The hand holding the toothbrush is steady, with fingers gripping the handle, indicating a controlled application. The surface appears smooth, with no visible resistance, allowing the substance to spread easily. The lighting is bright, casting a soft shadow of the toothbrush on the surface, enhancing the visibility of the application process.

**Q1:** Is the toothbrush handle described as plastic?
**Q2:** Did the paint cling to the bristles?
**Q3:** Was the paint stored in a can?
**Q4:** Was the canvas described as black?
**Q5:** Was the canvas backdrop described as vertical?
**Q6:** Did the person perform a flicking motion with the toothbrush?
**Q7:** Did droplets of paint detach from the bristles?
**Q8:** Was the detachment of paint caused by the force of the flick?
**Q9:** Did the paint droplets travel through the air?
**Q10:** Was gravity mentioned as influencing the droplets' travel?
**Q11:** Did the paint droplets adhere to the canvas surface?
**Q12:** Was the person's wrist motion described as controlled?
**Q13:** Did the person dip the bristles into the paint?
**Q14:** Was the paint described as thick?
**Q15:** Did the flicking motion require energy input from the person?

**Respond only as:**
Q1: Yes
Q2: No
...

# D  A TAXONOMY OF PHYSICALLY GROUNDED FAILURE MODES

To provide a more fine-grained analysis of *why* certain prompts are challenging for current T2V models, we constructed a structured annotation framework that captures deeper physical reasoning requirements beyond surface interaction types. Each prompt is characterized along three complementary axes:

**Action Complexity (AC).**

- **AC-1 (Direct):** A single, simple action.
- **AC-2 (Goal-Oriented):** A preparatory or two-step action aimed at achieving a specific result.
- **AC-3 (Procedural):** A multi-step, complex procedure where correct sequencing is essential.

**Scientific Principle (SP).**

- **SP-1 (Basic):** Straightforward mechanics with no salient principle.
- **SP-2 (Applied):** Application of a common, understandable physical principle (e.g., friction, leverage).
- **SP-3 (Complex/Hidden):** Relies on less obvious or complex, multi-stage principles (e.g., negative pressure, magnetism).

**Object Functionality (OF).**

- **OF-1 (Intended Use):** Object used for its designed purpose.
- **OF-2 (Repurposed):** Object employed as an improvised tool.
- **OF-3 (System Component):** Object functioning within a broader physical system.

**Illustrative Examples.**

- **Example 1 (AC-1, SP-1, OF-1):**
  "Hands pour a handful of mixed coins into the opening of an empty, clear plastic milk jug."
  → *Direct transfer using object as intended; straightforward use of objects.*
- **Example 2 (AC-1, SP-1, OF-2):**
  "A plastic bucket is placed upside down on someone's head like a hat."
  → *Simple physical manipulation but repurposed functionality.*
- **Example 3 (AC-1, SP-2, OF-3):**
  "A lit flashlight inside a translucent plastic bag makes the bag glow like a lantern."
  → *Requires reasoning about light diffusion and emergent system behavior.*
- **Example 4 (AC-2, SP-3, OF-1):**
  "A person melts a toy soldier with a magnifying glass focusing sunlight."
  → *Goal-driven use of a non-obvious scientific principle (solar heat focus).*
- **Example 5 (AC-3, SP-1, OF-1):**
  "Hands pour colorful powder into a small cup, press it down firmly, then invert the cup and tap it, releasing a shaped bath bomb onto a surface."
  → *Multi-step procedural action with material handling.*
- **Example 6 (AC-3, SP-3, OF-3):**
  "Hands squeeze an empty plastic water bottle, place the opening over an egg yolk, and release the squeeze, sucking the yolk into the bottle."
  → *Complex, multi-step use of pressure dynamics and system-level interaction.*

To apply this taxonomy at scale, we used Gemini 2.5 Pro to classify every prompt in our Medium, Hard, and Very Hard subsets. The resulting averages reported in Table 7 show a monotonic increase

Table 7: **Average taxonomy scores across difficulty tiers.** Higher difficulty subsets consistently show increased action complexity, more advanced scientific principles, and more complex object functionality. This confirms that PhysVidBench difficulty tiers reflect systematically deeper physical-reasoning demands rather than random noise.

| Difficulty Subset | AC | SP | OF | Overall Score |
|---|---|---|---|---|
| Medium Prompts | 1.48 | 1.65 | 1.62 | 4.75 |
| Hard Prompts | 1.69 | 1.77 | 1.90 | 5.36 |
| Very Hard Prompts | 2.06 | 1.94 | 2.10 | 6.10 |

in complexity along all three axes, confirming that higher-tier prompts demand more elaborate procedures, deeper physical principles, and more complex object functionality.

This structured analysis provides fine-grained diagnostic clarity: models often succeed on simple interactions (AC-1, SP-1, OF-1) but fail when tasks require multi-step procedures, non-obvious physical principles, or functional recontextualization. This taxonomy thus enhances PhysVidBench's value as a diagnostic tool—revealing not only how well models perform, but why they fail.

# E    AN END-TO-END EXAMPLE EVALUATION

---

**Prompt Generation**

**Stage 1. Input: Human Commonsense Goal + Solutions (PIQA)**

Goal: *Bag*
Solution: *can be used as a lantern with a flashlight indoors.*

- - - - - - - - - - - - - - - - - - - - - - - - - - - - - - - - - - - - - - - - - - - - - - - - - - - -

**Stage 2. Base Prompt:**

A lit flashlight inside a translucent plastic bag makes the bag glow like a lantern.

- - - - - - - - - - - - - - - - - - - - - - - - - - - - - - - - - - - - - - - - - - - - - - - - - - - -

**Stage 3. Upsmapled Prompt:**

A cylindrical flashlight, currently switched on and emitting a focused beam of bright light, is positioned entirely inside a thin, flexible, translucent plastic bag. The lightweight plastic material loosely conforms around the flashlight. Light rays emanating from the flashlight's bulb strike the inner walls of the bag; due to the plastic's translucent property, the light scatters and diffuses as it passes through, causing the entire volume of the bag to glow softly from within, resembling a makeshift lantern in a dimly lit space.

- - - - - - - - - - - - - - - - - - - - - - - - - - - - - - - - - - - - - - - - - - - - - - - - - - - -

---

**Evaluation**

**Stage 1. Physics-Grounded Question Generation:**

We generate questions that cover seven key physical commonsense dimensions defined in our ontology:

**AU:** Action & Procedural Understanding, **FM:** Force and Motion, **FP:** Fundamental Physics, **MT:** Material Interaction & Transformation, **OP:** Object Properties & Affordances, **SR:** Spatial Reasoning, **TD:** Temporal Dynamics.

| #   | Question                                                                  | Dimension(s) |
|-----|---------------------------------------------------------------------------|--------------|
| Q1  | Is the flashlight emitting light energy?                                  | FP           |
| Q2  | Is the bag made of plastic?                                               | OP           |
| Q3  | Is the flashlight entirely inside the bag?                                | SR, OP       |
| Q4  | Is the bag described as flexible?                                         | OP           |
| Q5  | Does the light scatter as it passes through the plastic?                  | MT, FP       |
| Q6  | Does the flashlight being on cause the bag to glow?                       | FP           |
| Q7  | Is the flashlight described as cylindrical?                               | SR, OP       |
| Q8  | Does the bag glow softly?                                                 | FP           |
| Q9  | Is the bag described as lightweight?                                      | OP           |
| Q10 | Does the setup resemble a makeshift lantern?                              | AU           |
| Q11 | Is the light emitted by the flashlight described as a focused beam initially? | FP, OP    |
| Q12 | Does the entire volume of the bag glow?                                   | SR, FP       |
| Q13 | Does the bag diffuse the light passing through it?                        | MT, FP       |

- - - - - - - - - - - - - - - - - - - - - - - - - - - - - - - - - - - - - - - - - - - - - - - - - - - -

**Stage 2. Dense Video Captioning:**

The generated video is captioned along eight physical commonsense dimensions:

**General:**
*The video features a close-up view of a bright light source, which appears to be a flashlight*

---

*or a similar handheld light. The light is emitting a strong, white beam that illuminates the surrounding area with a soft glow. The light is focused and directed, creating a halo effect that enhances the visibility of the surrounding environment. The background is blurred, but hints of blue and white colors suggest a cool, possibly outdoor setting. The light source is partially obscured by a translucent, plastic bag, which adds a layer of texture to the scene. The overall composition is centered on the light, drawing attention to its brightness and the way it interacts with the environment*

**Fundamental Physics (FP):**
*The video captures a moment where a light source, likely a bulb, is enveloped in a translucent material, creating a visual effect that resembles a flower. The energy transfer is evident as the light shines through the material, creating a soft glow that illuminates the surrounding area. The material appears to be in a state of tension, with the light source at the center, creating a dynamic interplay of light and shadow. The balance is slightly tilted, with the light source being the focal point, drawing the viewer's attention. The physical state of the material is fluid, allowing the light to pass through, which changes the overall appearance of the object from a solid to a more ethereal, glowing form.*

**Object Properties & Affordances (OP):**
*The main objects in the video are the plastic bag and the light bulb. The plastic bag, with its smooth and reflective surface, allows light to pass through, creating a soft glow. The light bulb, with its metallic base, emits a bright light that illuminates the bag. The actions that can be directly observed are the light bulb emitting light and the plastic bag reflecting and diffusing the light. The materials of the bag and bulb are visually confirmed as soft and metallic, respectively. The bulb's rigidity is suggested by its shape and the way it stands upright.*

**Spatial Reasoning (SR):**
*The video captures a close-up view of a light bulb, which is the main subject. The bulb is encased in a clear, plastic bag, creating a soft glow that illuminates the surrounding area. The bag appears to be slightly crumpled, adding texture to the visual. The light emanating from the bulb creates a bright, circular area that contrasts with the darker background. The bulb is positioned centrally in the frame, with the plastic bag slightly tilted, allowing the light to shine outward. The overall layout is simple, focusing on the interplay of light and shadow.*

**Temporal Dynamics (TD):**
*The video captures a moment where a bright light source, likely a flashlight, is being directed towards a translucent, reflective surface, creating a dynamic interplay of light and shadow. The light appears to be moving, as evidenced by the varying intensity of the illumination and the shifting patterns of light and dark. The surface reflects the light, creating a soft glow that enhances the visual effect. The overall atmosphere is ethereal and slightly blurred, with the focus primarily on the light source and its reflection. The light source is the main subject, with the surrounding environment remaining indistinct, allowing the viewer to concentrate on the interplay of light and shadow.*

**Action & Procedural Understanding (AU):**
*The video captures a moment where a bright light source, possibly a flashlight, is being directed towards a translucent, reflective surface, creating a dynamic interplay of light and shadow. The light source is the main focus, with its beam creating a luminous effect that illuminates the surrounding area. The reflective surface enhances the visual impact, as it captures and distorts the light, creating a sense of depth and movement. The agent's actions are deliberate, as they manipulate the light source to achieve the desired effect. The overall goal appears to be the exploration of light and reflection, showcasing the interplay between the light source and the reflective surface.*

**Material Interaction & Transformation (MT):**
*The video captures a moment where a clear plastic bag is being manipulated, likely by a hand, as it is folded and twisted. The bag's transparency allows the viewer to see the light source within, creating a bright, radiant effect. The light source appears to be the main focus, with its*

*glow illuminating the bag and the surrounding environment. The bag's material, while initially clear, shows signs of distortion, possibly due to the folding and twisting actions. The light creates a soft glow that reflects off the bag's surface, enhancing the visual appeal of the scene. The overall interaction between the bag and the light source is one of visual interest, with the bag's transparency and the light's brightness creating a striking contrast.*

**Force and Motion (FM):**
*The video captures a moment where a clear plastic bag is being manipulated, likely by a hand, as evidenced by the slight blur around the edges of the bag. The bag is being stretched and folded, creating a dynamic interplay of light and shadow. The bright light source creates a lens flare effect, which adds to the visual complexity. The bag's transparency allows for a view of the light source, which is bright and white, indicating a strong light source. The bag's surface reflects the light, creating a shimmering effect. The overall scene is one of gentle interaction, with the bag responding to the touch of the hand, creating a fluid motion that is both delicate and deliberate.*

- - - - - - - - - - - - - - - - - - - - - - - - - - - - - - - - - - - - - - - - - - - - - - - - -

**Stage 3. QA-Based Evaluation (LLM-as-a-Judge)**

| # | Question | Answer |
|---|---|---|
| Q1 | Is the flashlight emitting light energy? | Yes |
| Q2 | Is the bag made of plastic? | Yes |
| Q3 | Is the flashlight entirely inside the bag? | Yes |
| Q4 | Is the bag described as flexible? | Yes |
| Q5 | Does the light scatter as it passes through the plastic? | Yes |
| Q6 | Does the flashlight being on cause the bag to glow? | Yes |
| Q7 | Is the flashlight described as cylindrical? | No |
| Q8 | Does the bag glow softly? | Yes |
| Q9 | Is the bag described as lightweight? | No |
| Q10 | Does the setup resemble a makeshift lantern? | No |
| Q11 | Is the light emitted by the flashlight described as a focused beam initially? | Yes |
| Q12 | Does the entire volume of the bag glow? | No |
| Q13 | Does the bag diffuse the light passing through it? | Yes |

# F ADDITIONAL GENERATION RESULTS AND QUALITATIVE EXAMPLES

## F.1 QUALITATIVE EXAMPLES

To illustrate the behaviors and failure modes of current text-to-video models on PhysVidBench, we present several representative qualitative cases. For each case, we show: (1) the input text prompt, (2) key frames from videos generated by two contrasting models, and (3) a subset of yes/no QA outcomes used for scoring. *Note that only four questions are shown per example for brevity; the full evaluation uses a broader set of questions covering multiple physical-commonsense dimensions.* These examples highlight where models succeed or break down across different reasoning skills.

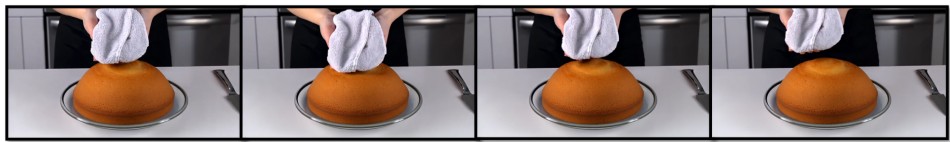

**Prompt:** Hands place a tea towel over a freshly baked cake with a domed top and gently press down, flattening the dome.

---

**Prompt-based Evaluation Questions**

**Model: Cosmos (14B)**

**Q1:** Does the fabric drape smoothly over the cake's dome?
**A1:** Yes

**Q2:** Is the tea towel described as rectangular?
**A2:** No

**Q3:** Does the cake visibly flatten as a result of the pressure?
**A3:** Yes

**Q4:** Does the warmth of the cake contribute to its structure yielding under pressure?
**A4:** No

---

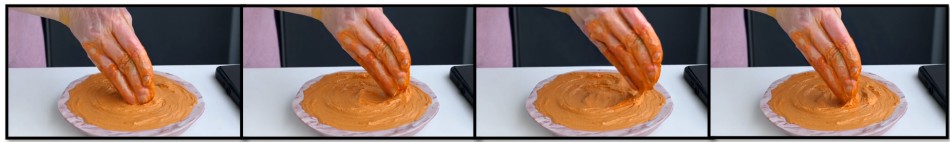

**Prompt:** Hands mix baking soda and water in a small bowl to form a paste, then scoop some paste and rub it onto skin with orange streaks.

---

**Prompt-based Evaluation Questions**

**Model: Cosmos (14B)**

**Q1:** Did fingers mix the contents of the bowl?
**A1:** Yes

**Q2:** Was the resulting paste described as opaque?
**A2:** No

**Q3:** Were there orange streaks on the surface where the paste was applied?
**A3:** No

**Q4:** Was the bowl used for containment of the ingredients?
**A4:** Yes

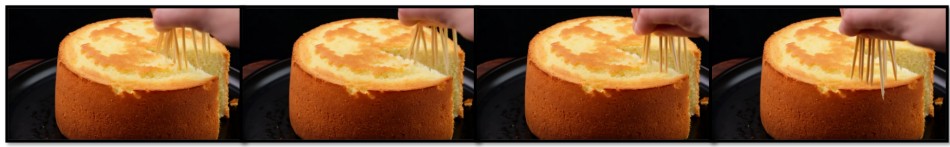

**Prompt:** Toothpicks are pushed through slices of bread, securing them against the cut edges of a leftover cake.

---

**Prompt-based Evaluation Questions**

**Model: Hunyuan**

**Q1:** Are the toothpicks made of wood?
**A1:** Yes

**Q2:** Are the toothpicks held firmly?
**A2:** No

**Q3:** Do the toothpicks embed themselves within the cake's interior?
**A3:** Yes

**Q4:** Is the bread slice secured against the cake by the action?
**A4:** Yes

---

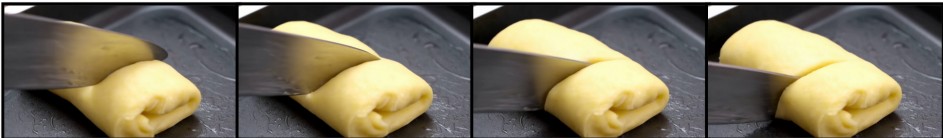

**Prompt:** The edge of a metal spoon presses down into a roll of soft potato dough, cutting off a small piece.

---

**Prompt-based Evaluation Questions**

**Model: Hunyuan**

**Q1:** Does the action result in separating a portion of the dough?
**A1:** Yes

**Q2:** Is the dough described as having a cylindrical shape?
**A2:** Yes

**Q3:** Does the spoon shear through the dough's structure?
**A3:** No

**Q4:** Is the spoon described as being made of metal?
**A4:** No

---

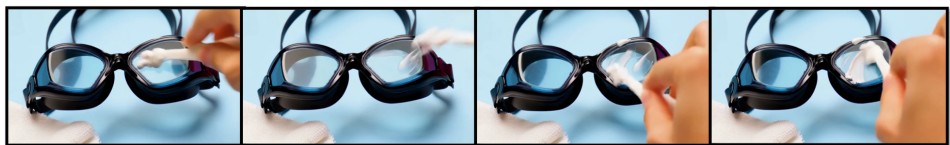

**Prompt:** Hands stuff dryer lint into cardboard toilet paper tubes.

| Prompt-based Evaluation Questions |
|---|

**Model: Wan2.1 (14B)**

**Q1:** Is the tube held steady by one hand?
**A1:** No

**Q2:** Are two hands involved in the overall action described?
**A2:** Yes

**Q3:** Does the volume occupied by the lint decrease when pushed into the tube?
**A3:** Yes

**Q4:** Is force applied by fingers to push the lint?
**A4:** Yes

**Prompt:** Toothpaste is smeared onto the inner lens of swimming goggles and then wiped away with a soft cloth.

| Prompt-based Evaluation Questions |
|---|

**Model: Wan2.1 (14B)**

**Q1:** Did the toothpaste adhere to the lens surface after application?
**A1:** Yes

**Q2:** Was the cloth pressed against the lens?
**A2:** Yes

**Q3:** Did the wiping action remove the toothpaste from the lens?
**A3:** No

**Q4:** Is the goggle lens described as hard?
**A4:** No

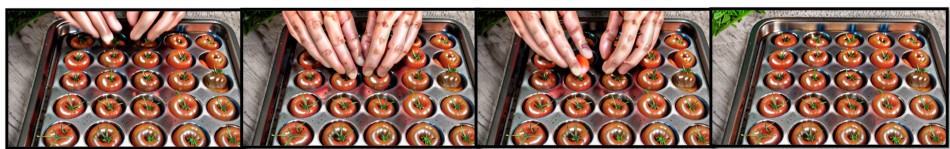

**Prompt:** Hands place whole cherry tomatoes into the individual cups of a greased metal muffin tin.

**Prompt-based Evaluation Questions**

**Model: LTX-Video**

**Q1:** Is the muffin tin made of metal?
**A1:** Yes

**Q2:** Are the tomatoes described as small and round?
**A2:** No

**Q3:** Does gravity cause the released tomato to move downwards?
**A3:** No

**Q4:** Are the tomatoes placed into the cups one at a time?
**A4:** Yes

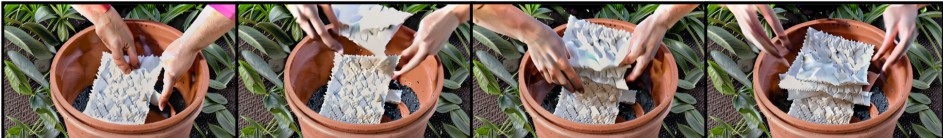

**Prompt:** Hands place several sheets of white styrofoam into the bottom of a terracotta planter.

**Prompt-based Evaluation Questions**

**Model: LTX-Video**

**Q1:** Do the hands lower the sheets into the planter?
**A1:** Yes

**Q2:** Do the sheets come to rest on the bottom surface of the planter?
**A2:** No

**Q3:** Is the styrofoam material described as porous?
**A3:** No

**Q4:** Is the planter described as sturdy?
**A4:** No

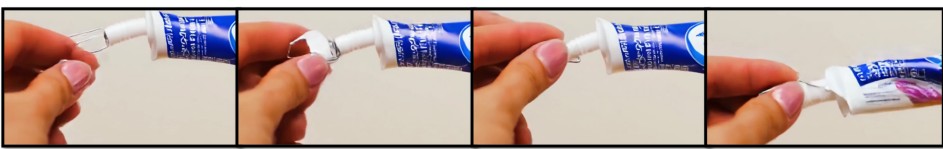

**Prompt:** A person attaches a binder clip to the end of a toothpaste tube and squeezes it to get more toothpaste out.

| Prompt-based Evaluation Questions |
| --- |

**Model: Wan2.1 (1.3B)**

**Q1:** Does the clip's spring mechanism compress the tube?
**A1:** No

**Q2:** Is the toothpaste described as viscous?
**A2:** No

**Q3:** Does the binder clip create a barrier preventing toothpaste from moving back towards the flattened end?
**A3:** No

**Q4:** Does the clip help to gather the remaining toothpaste near the opening?
**A4:** No

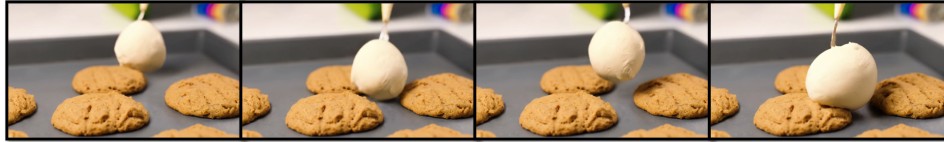

**Prompt:** An ice cream scoop digs into cookie dough and releases a perfect ball onto a baking sheet.

| Prompt-based Evaluation Questions |
| --- |

**Model: Wan2.1 (1.3B)**

**Q1:** Is the cookie dough described as soft?
**A1:** Yes

**Q2:** Does the dough yield under pressure from the scoop?
**A2:** No

**Q3:** Is the scoop positioned directly above the baking sheet before the dough is released?
**A3:** No

**Q4:** Is the ejected dough ball described as spherical?
**A4:** No

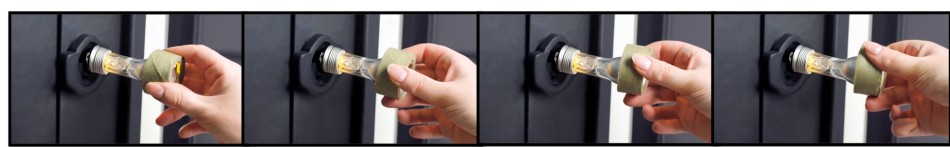

**Prompt:** Duct tape is wrapped around a stuck light bulb in a socket, and a hand uses the tape to twist the bulb counter-clockwise, loosening it.

| Prompt-based Evaluation Questions |
| --- |

**Model: Cosmos (7B)**

**Q1:** Is the tape described as having a flexible fabric backing?
**A1:** No

**Q2:** Is the tape wrapped multiple times around the bulb?
**A2:** No

**Q3:** Does a hand grasp the layered duct tape on the bulb?
**A3:** No

**Q4:** Does the hand twist in a counter-clockwise direction?
**A4:** No

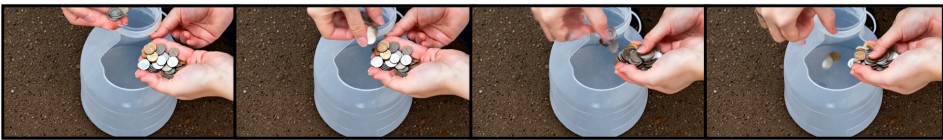

**Prompt:** Hands pour a handful of mixed coins into the opening of an empty, clear plastic milk jug.

| Prompt-based Evaluation Questions |
| --- |

**Model: Cosmos (7B)**

**Q1:** Does the action of tilting the hands happen before the coins begin to fall?
**A1:** No

**Q2:** Does gravity influence the movement of the coins after release?
**A2:** No

**Q3:** Do the coins pass completely through the jug's opening to enter the body?
**A3:** No

**Q4:** Are the coins contained within the hands before being released?
**A4:** No

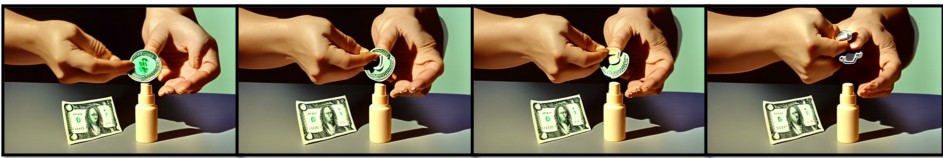

**Prompt:** Hands place folded dryer sheets inside a pair of sneakers.

| Prompt-based Evaluation Questions |
| --- |

**Model: VideoCrafter2**

**Q1:** Do the hands push a dryer sheet into the first sneaker?
**A1:** No

**Q2:** Is the action of inserting a dryer sheet performed on both sneakers mentioned in the prompt?
**A2:** No

**Q3:** Is the dryer sheet described as flexible?
**A3:** No

**Q4:** Is the dryer sheet inserted into a cavity within the sneaker?
**A4:** No

**Prompt:** Hands place keys and folded money into a hollowed-out suntan lotion bottle and put the top back on.

| Prompt-based Evaluation Questions |
| --- |

**Model: VideoCrafter2**

**Q1:** Are the keys described as rigid?
**A1:** No

**Q2:** Is the bottle described as hollowed-out?
**A2:** Yes

**Q3:** Are the keys inserted through an opening in the bottle?
**A3:** No

**Q4:** Is the bottle made of plastic?
**A4:** Yes

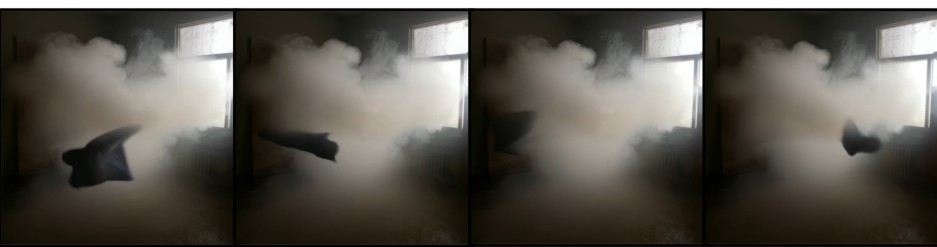

**Prompt:** Ice cubes are poured into a kitchen sink drain and ground up by the garbage disposal.

> ### Prompt-based Evaluation Questions
>
> **Model: MAGI-1**
>
> **Q1:** Do the cubes tumble downwards into a cavity?
> **A1:** No
>
> **Q2:** Is the disposal chamber described as a confined space?
> **A2:** No
>
> **Q3:** Do the spinning components collide forcefully with the ice cubes?
> **A3:** No
>
> **Q4:** Do the internal components of the disposal spin rapidly?
> **A4:** No

**Prompt:** A damp towel is spun rapidly in a room filled with visible smoke, causing the smoke to swirl and dissipate slightly.

> ### Prompt-based Evaluation Questions
>
> **Model: MAGI-1**
>
> **Q1:** Was the air in the room initially still before the spinning?
> **A1:** Yes
>
> **Q2:** Does the damp surface of the towel interact with the smoke particles?
> **A2:** No
>
> **Q3:** Is the towel's motion described as being driven against air resistance?
> **A3:** Yes
>
> **Q4:** Does the rapid movement of the towel generate air currents?
> **A4:** Yes

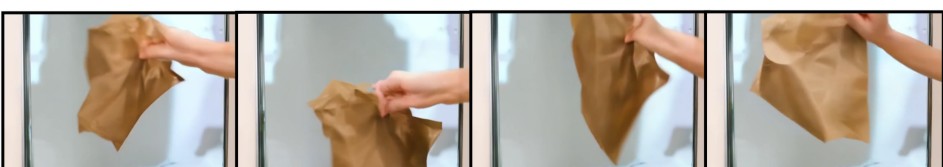

**Prompt:** A hand wipes a glass mirror clean using a crumpled brown paper bag.

### Prompt-based Evaluation Questions

**Model: CogVideoX (2B)**

**Q1:** Is the paper bag described as flexible?
**A1:** Yes

**Q2:** Does the manipulation of the bag into a wad happen *before* it is pressed against the mirror?
**A2:** No

**Q3:** Is the described action a wiping motion?
**A3:** No

**Q4:** Is pressure applied by the hand onto the paper wad against the mirror?
**A4:** No

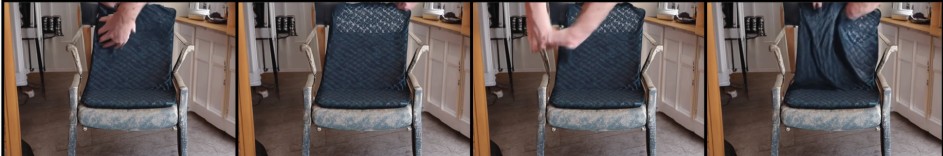

**Prompt:** A decorative automobile bucket seat cover is stretched and fitted over an old, worn desk chair.

### Prompt-based Evaluation Questions

**Model: CogVideoX (2B)**

**Q1:** Was the cover pulled over the backrest first?
**A1:** No

**Q2:** Did the application process involve stretching the cover?
**A2:** Yes

**Q3:** Does the cover conform closely to the chair's shape?
**A3:** Yes

**Q4:** Does the chair have underlying padding mentioned?
**A4:** No

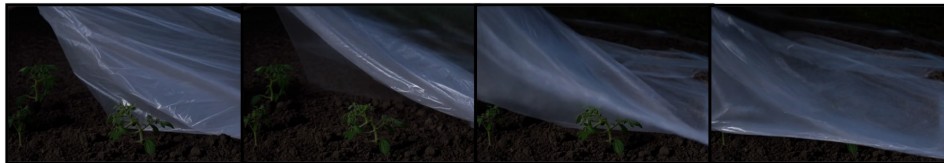

**Prompt:** Clear plastic sheeting is carefully draped over several small tomato plants in a garden bed at dusk.

**Prompt-based Evaluation Questions**

**Model: CogVideoX (5B)**

**Q1:** Does gravity cause the plastic sheet to move downwards?
**A1:** No

**Q2:** Does the plastic sheet change shape as it settles on the plants?
**A2:** Yes

**Q3:** Does the plastic sheet cover the tomato plants?
**A3:** Yes

**Q4:** Does the flexibility of the plastic allow it to bend over the plants?
**A4:** Yes

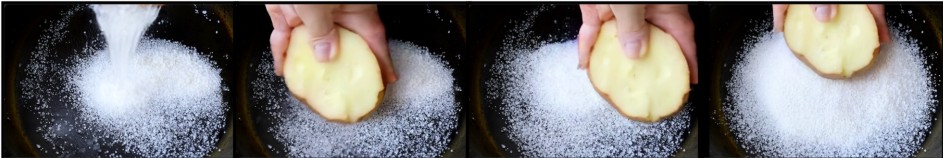

**Prompt:** Coarse salt is poured into a dirty cast iron skillet, then scrubbed vigorously with the cut side of a potato half.

**Prompt-based Evaluation Questions**

**Model: CogVideoX (5B)**

**Q1:** Does the hand apply pressure while rubbing the potato?
**A1:** No

**Q2:** Is the salt ground between the potato and the skillet surface?
**A2:** No

**Q3:** Is the concrete contained within the mold?
**A3:** Yes

**Q4:** Is the salt contained within the skillet's interior after pouring?
**A4:** No

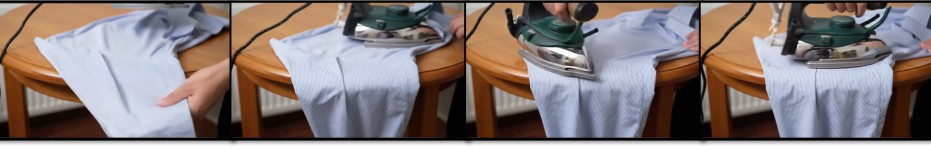

**Prompt:** A shirt sleeve draped over the edge of a wooden table is being pressed flat by a moving iron.

Prompt-based Evaluation Questions

**Model: Sora**

**Q1:** Does the iron apply downward pressure onto the fabric?
**A1:** Yes

**Q2:** Does the iron transfer heat to the fabric?
**A2:** Yes

**Q3:** Is the soleplate of the iron made of metal?
**A3:** Yes

**Q4:** Is the soleplate of the iron flat?
**A4:** No

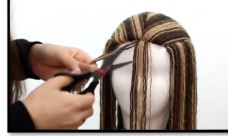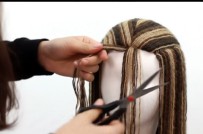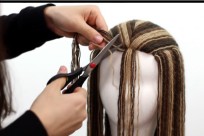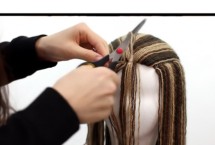

**Prompt:** Hands cut strands of yarn with scissors, then apply glue to the ends and press them onto a mannequin head.

Prompt-based Evaluation Questions

**Model: Sora**

**Q1:** Does squeezing the scissor handles cause the blades to pivot?
**A1:** Yes

**Q2:** Is the adhesive described as a liquid when applied?
**A2:** No

**Q3:** Are multiple strands of yarn held by the hands initially?
**A3:** No

**Q4:** Does the cutting action result in loose strands?
**A4:** Yes

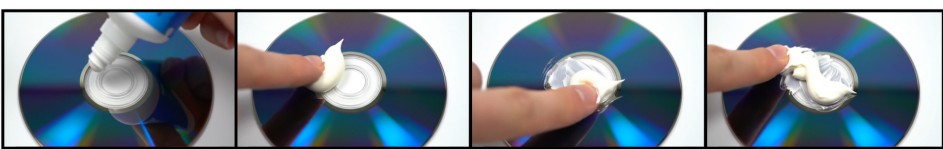

**Prompt:** Toothpaste is squeezed onto a scratched CD, then rubbed radially outwards with a finger.

| Prompt-based Evaluation Questions |
|---|

**Model: Veo-2**

**Q1:** Is the finger used to apply force to spread the paste?
**A1:** Yes

**Q2:** Is the CD's state of rest maintained throughout the toothpaste application and spreading?
**A2:** No

**Q3:** Does the rubbing action cause the toothpaste to cover the scratched area?
**A3:** No

**Q4:** Is the CD described as rigid?
**A4:** Yes

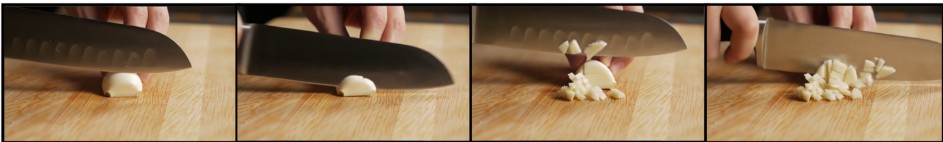

**Prompt:** The flat side of a chef's knife is pressed down onto a garlic clove on a cutting board, crushing it.

| Prompt-based Evaluation Questions |
|---|

**Model: Veo-2**

**Q1:** Does the garlic clove audibly crack?
**A1:** No

**Q2:** Is downward force applied to the knife?
**A2:** Yes

**Q3:** Is the knife described as rigid?
**A3:** Yes

**Q4:** Is the knife positioned parallel to the cutting board surface?
**A4:** No

## F.2 MODEL GENERATION SPECIFICATIONS

Table 8 summarizes, for each evaluated text-to-video (T2V) model, the output resolution, number of frames, video duration and guidance scale used throughout our experiments. We use these settings consistently when generating both base and upsampled videos to ensure a fair comparison across models.

Table 8: **Model generation specifications.** This table lists the generation settings for each text-to-video (T2V) model used in our experiments. For each model, we report the video resolution (in width and height), number of frames, total video duration (in seconds), and the guidance scale used during sampling (if applicable). These settings reflect the default or recommended configurations for each model and were held consistent across evaluations to ensure comparability in visual output and downstream analysis.

| Model | Resolution (w/h) | # of Frames | Video Duration | Guidance Scale |
|---|---|---|---|---|
| LTX-Video | 1216x704 | 121 | 4 | 3 |
| VideoCrafter2 | 512x320 | 16 | 1 | 12 |
| CogVideoX (2B) | 720x480 | 49 | 6 | 6 |
| CogVideoX (5B) | 720x480 | 49 | 6 | 6 |
| Wan2.1 (1.3B) | 832x480 | 33 | 6 | 5 |
| Wan2.1 (14B) | 832x480 | 33 | 6 | 5 |
| MAGI-1 | 720x720 | 48 | 2 | 7.5 |
| Hunyuan Video | 960x544 | 129 | 5 | 6 |
| Cosmos (7B) | 1280x704 | 121 | 5 | 7 |
| Cosmos (14B) | 1280x704 | 121 | 5 | 7 |
| Sora | 1280x720 | 150 | 5 | - |
| Veo-2 | 1280x720 | 120 | 5 | - |

## F.3 EFFECT OF CAPTION DETAIL ON MODEL PERFORMANCE: SHORT VS. DENSE CAPTIONS

The level of detail in captions significantly affects the accuracy and reliability of evaluations conducted using vision-language models (VLMs). Short captions tend to provide high-level summaries and often omit subtle yet critical details, such as precise object interactions or slight motions, leading to incomplete or inaccurate physical commonsense judgments. Moreover, when directly queried, VLMs may hallucinate or exaggerate minor visual cues, further degrading evaluation quality.

Dense captioning, by contrast, captures fine-grained observations without suggestive prompting. We employ AuroraCap for its proven ability to generate detailed, multi-facet descriptions. Specifically, for each video we produce one general-purpose caption plus seven dimension-specific captions aligned with our physical commonsense ontology (e.g., spatial layout, force dynamics, material behavior). This explicit, targeted prompting ensures comprehensive coverage of all relevant visual cues.

During evaluation, a yes/no question is marked correct if any of the eight captions supplies the required evidence. By pooling information across multiple, detailed captions, we drastically reduce errors from missing details or hallucinations. This approach highlights the clear advantage of dense, dimension-targeted captioning over short, generic summaries in assessing physical commonsense.

## G   USER STUDY DETAILS AND HUMAN EVALUATION

To verify that our automatic, caption-based evaluation (auto-eval pipeline) aligns with human judgment, we carried out a user study via Qualtrics. We selected the four top-performing open-source models, namely Cosmos-14B, WanAI-2.1 (14B), MAGI-1, and Hunyuan Video, and sampled 30 prompts (out of 383 upsampled) at random. Each prompt was used to generate one video per model, yielding 120 videos in total.

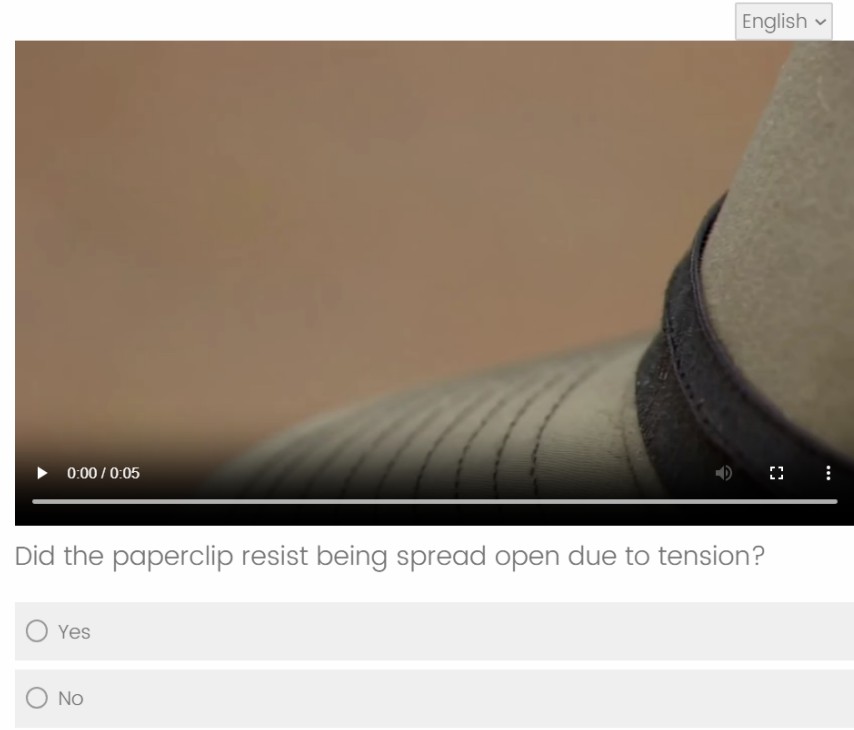

Figure 6: **User Evaluation Interface.** Our web-based interface for the human study mimicked the auto-eval QA structure. Participants watched a video, then answered five Yes/No questions drawn from the same QA bank used in automatic evaluation. The design enforced randomization across videos and questions, while masking model and prompt identity to reduce bias.

**Procedure.** For each video, we used the yes/no questions produced by our auto-eval pipeline and presented them in a survey interface built with Qualtrics that mimics the automated QA layout (Fig. 6). Participants saw each video alongside five randomly assigned questions, answered in a Yes/No format. Videos and questions were fully randomized to prevent any inference of prompt or model identity. Before beginning, participants read brief instructions on interpreting the questions; we then collected non-identifying metadata (age range and self-reported familiarity with deep learning).

**Participants.** Fifteen individuals took part, collectively providing 1,340 binary (Yes/No) responses across the 120 videos. Owing to the randomized assignment and modest pool size, some questions were answered only once, while others received multiple responses.

**Results and Analysis.** We computed Pearson correlation coefficients to assess alignment between auto-eval scores and human judgments. Consistent with expectations, correlation increased as more annotators responded: for the MAGI-1 model, questions answered by only one participant yielded a correlation of $r=0.47$, while those answered by at least three participants reached $r=0.69$. Fig. 7 plots these correlations (with linear fits) for MAGI-1, Cosmos, Hunyuan, and WanAI, demonstrating varying degrees of agreement.

**Summary.** These findings confirm that our caption-based LLM evaluation closely tracks collective human judgment, and that increasing the number of annotators further strengthens this alignment.

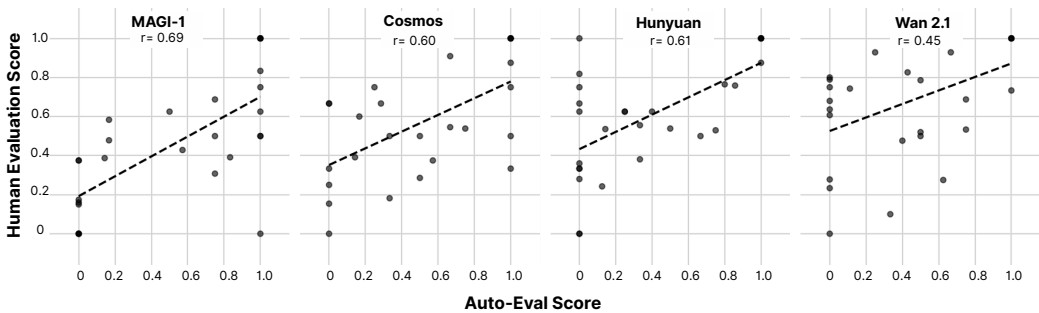

Figure 7: **Pearson Correlation between Human Evaluations and Auto-Evaluations.** Plots comparing auto-eval scores with human judgment scores across 30 prompts for four open-source top-performing models. Each subplot shows a linear fit and corresponding Pearson correlation coefficient ($r$): MAGI-1 ($r = 0.60$), Cosmos ($r = 0.69$), Hunyuan ($r = 0.61$), and Wan 2.1 ($r = 0.45$). Results demonstrate varying degrees of alignment between automated and human assessments.

## H ROBUSTNESS OF THE AUTOMATIC EVALUATION PIPELINE

Our automatic, caption–based evaluation ("auto–eval") was designed and validated with robustness as a primary objective. Below we compile the empirical evidence supporting its reliability and the analyses we performed to stress–test each component.

### H.1 DESIGN PRINCIPLES FOR ROBUSTNESS

**Decoupled perception and reasoning.** To avoid end–to–end failure compounding in direct video–question answering, we *separate* perception (dense, multi–perspective captioning) from reasoning (LLM QA over text). This modularity makes error sources observable and debuggable.

**Redundant multi–perspective evidence.** Each video is captioned 8 times (one general + seven captions targeted to our physical ontology dimensions), increasing recall of relevant cues and reducing the influence of any single caption omission.

**Dense, dimension–balanced probing.** On average $\sim 11$ physics–grounded yes/no questions are posed per video, distributed across dimensions (forces, materials, spatial relations, procedures, etc.). Final scores are aggregated over questions and dimensions, lowering variance from any individual item.

### H.2 CROSS–JUDGE STABILITY

To test dependence on a particular language model, we swapped the judge from Gemini–Flash to DeepSeek. The induced ranking of T2V models is highly stable (Spearman $\rho$=0.98 across full model lists), indicating the pipeline captures differences in *videos*, not idiosyncrasies of the judge (Table 9).

| Model | Gemini–Flash Rank | DeepSeek Rank |
|---|---|---|
| Cosmos (14B) | 1 | 1 |
| Cosmos (7B) | 2 | 3 |
| Wan2.1 (14B) | 3 | 2 |
| MAGI–1 | 4 | 4 |
| Wan2.1 (1.3B) | 5 | 5 |
| Hunyuan Video | 6 | 6 |
| CogVideoX (2B) | 7 | 7 |
| VideoCrafter2 | 8 | 9 |
| CogVideoX (5B) | 9 | 8 |
| LTX–Video | 10 | 10 |
| | | Spearman $\rho = 0.98$ (rank stability) |

Table 9: Cross–judge stability of our evaluation pipeline. Replacing Gemini–Flash with DeepSeek yields nearly identical model rankings, indicating the performance signal is not judge–specific but emerges from our multi–perspective, question–driven evaluation design.

### H.3 CAPTIONER ANALYSIS AND CROSS–CAPTIONER STABILITY

To assess whether PhysVidBench depends on a particular captioning model, we re-ran the entire evaluation pipeline using two alternative captioners, Qwen-2.5-VL-7B and Qwen-3-VL-30B, in place of AuroraCap. Across all ten T2V models, we observe strong cross-captioner agreement, The Spearman rank correlations between captioners range from 0.80–0.98, indicating that relative ordering of T2V models is effectively unchanged across captioners. Notably, Cosmos-14B, Wan2.1-14B, MAGI-1, and Hunyuan Video consistently appear as the top-performing models under all captioning variants, showing that the evaluator captures properties of the videos, not idiosyncrasies of any captioner. Pearson correlations of 0.74–0.98 between captioners confirm that absolute scores move coherently when switching captioners. Table 10 summarizes these results.

These results show that PhysVidBench is not captioner-dependent: swapping in significantly different captioning, including models with different vision–language backbones and training corpora, yields essentially the same model rankings and similar absolute scores. Combined with our multi-caption

Table 10: **Cross–captioner agreement.** Rank and Pearson correlations between AuroraCap, Qwen-2.5-VL-7B, and Qwen-3-VL-30B computed over model-level accuracies. High correlations demonstrate that PhysVidBench scores are stable across captioning models.

| Rank Correlation | AuroraCap | Qwen-2.5-VL-7B | Qwen-3-VL-30B |
|---|---|---|---|
| AuroraCap | 1 | 0.8 | 0.8 |
| Qwen-2.5-VL-7B | 0.8 | 1 | 1 |
| Qwen-3-VL-30B | 0.8 | 1 | 1 |

| Pearson Correlation | AuroraCap | Qwen-2.5-VL-7B | Qwen-3-VL-30B |
|---|---|---|---|
| AuroraCap | 1 | 0.83 | 0.74 |
| Qwen-2.5-VL-7B | 0.83 | 1 | 0.98 |
| Qwen-3-VL-30B | 0.74 | 0.98 | 1 |

design (one general and seven dimension-specific captions), this ensures that the evaluator is robust to individual caption omissions and reliably captures the physical content encoded in the video.

### H.4 SENSITIVITY TO QUESTION QUALITY

We manually audited and filtered $\sim 5\%$ problematic questions (e.g., referencing non–visual cues like sound, or prompting details present only in the upsampled prompt but not visually verifiable). Recomputing results with/without these items yields negligible changes to scores and *no* changes to model ranking (cf. the "Base (Filtered)" and "Upsampled (Filtered)" columns reported in our rebuttal). This shows the evaluation is robust to small fractions of imperfect items.

### H.5 RESISTANCE TO RESPONSE COLLAPSE AND HALLUCINATION

Direct use of VLMs in the evaluation underperform our method when measured against human judgments, and strong open VLMs (e.g., Video–LLaVA, InternVL3) frequently *collapse* to trivial "Yes" responses, yielding non–informative correlations (Table 5). By first converting video to text with a high–quality captioner and then reasoning over structured evidence, our pipeline avoids this mode collapse and achieves substantially higher human alignment.

### H.6 DISTINCTION FROM OTHER AUTOMATIC EVALUATION METRICS

Evaluation metrics optimized for generic visual quality or motion dynamics (VideoScore; VideoPhy–2 AutoEval) exhibit weak or negative correlation with human judgments on our *physical commonsense* benchmark. This divergence demonstrates that our benchmark probes a distinct capability (tool–mediated, causal physical reasoning) and that our evaluator is calibrated to that target, not to aesthetics or generic dynamics.

### H.7 UTILITY ROBUSTNESS: DEVELOPMENT LOOP IMPROVES MODELS

Beyond passive scoring, we used auto–eval as a diagnostic signal in a training–free self–refinement loop: detect failed dimensions $\rightarrow$ prompt rewrite $\rightarrow$ regenerate $\rightarrow$ re–evaluate. On 100 prompts, CogVideoX–2B and CogVideoX–5B improved from $21.6 \rightarrow 32.7$ (+29.8%) and $17.8 \rightarrow 29.7$ (+38.1%), respectively. The same evaluator that measures performance *also* guides improvement, validating its stability and usefulness for development.

---

**An Illustrative Example**

The example below shows how PhysVidBench feedback guided a revised prompt that led to a visibly more plausible generation:

**Base Prompt:** *A screwdriver tip is placed on the lid of a metal can, and a hammer taps the screwdriver handle, puncturing the lid.*

---

> **Refined Prompt (Guided by Feedback):** *A person grips a hammer, delivering a sharp, controlled tap to the rigid upper end of a vertically positioned screwdriver with a solid shaft; the heavy hammer's impact force travels down the screwdriver's shaft, concentrating at its tip, which is pressed firmly against the surface of a sealed metal can lid, causing the thin metal to yield and puncture after the hammer tap, overcoming the lid's structural integrity.*

Taken together, these analyses show that our evaluation pipeline (i) aligns strongly with human judgments, (ii) yields stable rankings across different LLM judges, (iii) is insensitive to small fractions of imperfect questions, (iv) avoids the collapse modes observed in direct VLM-QA, (v) measures a distinct capability not captured by generic video metrics, and (vi) can serve as a practical diagnostic tool to improve T2V outputs. These properties make it a robust and scalable proxy for human evaluation in physical commonsense video generation.

# I   STATISTICAL SIGNIFICANCE AND CONFIDENCE INTERVAL ANALYSIS

Across all analyses, the bootstrap confidence intervals provide clear evidence that PhysVidBench yields stable and statistically meaningful evaluation signals. Using 10k bootstrap resamples, we find that for most models, the confidence intervals on the accuracy difference between upsampled and base prompts lie entirely above zero, indicating statistically significant gains from prompt enrichment. The remaining models have intervals that only lightly touch zero, with lower bounds extremely close to zero and consistently positive mean differences. As shown in Table 11, these patterns hold uniformly across all evaluated models, suggesting that the observed improvements are systematic rather than noise-driven, even when effect sizes are modest.

Table 11: **Bootstrap 95% confidence intervals for model accuracies using upsampled prompts, base prompts, and their paired differences.** Positive intervals in the "Difference" column indicate statistically significant improvements from prompt enrichment.

| Model | Upsampled Prompts CI | Base Prompts CI | Difference |
|---|---|---|---|
| CogVideoX (2B) | [0.2341, 0.2776] | [0.2048, 0.2453] | [0.0060, 0.0565] |
| CogVideoX (5B) | [0.1878, 0.2320] | [0.1967, 0.1924] | [-0.0089, 0.0396] |
| WanAI-1.3B | [0.2826, 0.3304] | [0.2749, 0.2747] | [0.0077, 0.0557] |
| Wan2.1 (14B) | [0.3165, 0.3662] | [0.3251, 0.3213] | [-0.0086, 0.0449] |
| VideoCrafter2 | [0.2014, 0.2407] | [0.2027, 0.2050] | [-0.0013, 0.0357] |
| LTX-Video | [0.1886, 0.2325] | [0.1820, 0.1791] | [0.0066, 0.0534] |
| Cosmos (7B) | [0.3249, 0.3765] | [0.2694, 0.2651] | [0.0555, 0.1114] |
| Cosmos (14B) | [0.3396, 0.3846] | [0.2930, 0.2886] | [0.0466, 0.0960] |
| Hunyuan Video | [0.2764, 0.3249] | [0.2930, 0.2894] | [-0.0166, 0.0355] |
| MAGI-1 | [0.3010, 0.3508] | [0.2729, 0.2717] | [0.0281, 0.0791] |

For difficulty-based subsets, the bootstrap confidence intervals for Medium, Hard, and Very Hard prompts (Tables 12 and 13) show no overlap across tiers for any model, under both base and upsampled settings. This consistent separation indicates that the three subsets correspond to genuinely different levels of physical-reasoning complexity rather than noise. As expected, accuracy decreases as difficulty increases, and the decline is statistically significant across all models. These results validate the structured difficulty design of PhysVidBench and confirm that the benchmark reliably distinguishes model behavior across increasing levels of difficulty.

Table 12: **Bootstrap 95% confidence intervals across difficulty tiers for upsampled prompts.** Each interval reflects the model's accuracy uncertainty on the Very Hard, Hard, and Medium subsets. The clear non-overlap between tiers across all models demonstrates that PhysVidBench's difficulty stratification captures meaningful differences in physical-reasoning complexity.

| Model | Very Hard Prompts CI | Hard Prompts CI | Medium Prompts CI |
|---|---|---|---|
| CogVideoX (2B) | [0.0961, 0.1365] | [0.2551, 0.3087] | [0.4429, 0.5555] |
| CogVideoX (5B) | [0.0897, 0.1382] | [0.1934, 0.2481] | [0.3248, 0.4596] |
| Wan2.1 (1.3B) | [0.1227, 0.1697] | [0.3103, 0.3663] | [0.5210, 0.6285] |
| Wan2.1 (14B) | [0.1547, 0.2060] | [0.3419, 0.3998] | [0.5645, 0.6642] |
| VideoCrafter2 | [0.0876, 0.1275] | [0.2074, 0.2553] | [0.3958, 0.4938] |
| LTX-Video | [0.0729, 0.1110] | [0.2009, 0.2573] | [0.3489, 0.4742] |
| Cosmos (7B) | [0.1552, 0.2182] | [0.3574, 0.4186] | [0.5488, 0.6606] |
| Cosmos (14B) | [0.1951, 0.2487] | [0.3651, 0.4201] | [0.5528, 0.6319] |
| Hunyuan Video | [0.1336, 0.1859] | [0.3056, 0.3646] | [0.4667, 0.5830] |
| MAGI-1 | [0.1447, 0.1911] | [0.3229, 0.3830] | [0.5483, 0.6589] |

Table 13: **Bootstrap 95% confidence intervals across difficulty tiers for base prompts.** The monotonic separation across tiers holds for base prompts as well, indicating that the difficulty structure is robust even without enriched physical detail.

| Model | Very Hard Prompts CI | Hard Prompts CI | Medium Prompts CI |
|---|---|---|---|
| CogVideoX-2B | [0.0901, 0.1292] | [0.2251, 0.2775] | [0.3619, 0.4806] |
| CogVideoX-5B | [0.0710, 0.1151] | [0.1856, 0.2419] | [0.3026, 0.4287] |
| WanAI-1.3B | [0.1110, 0.1577] | [0.2650, 0.3189] | [0.4849, 0.5849] |
| WanAI-14B | [0.1543, 0.2127] | [0.3140, 0.3729] | [0.5065, 0.6238] |
| VideoCrafter2 | [0.0729, 0.1105] | [0.1949, 0.2448] | [0.3501, 0.4544] |
| LTX Video | [0.0604, 0.0979] | [0.1601, 0.2083] | [0.3358, 0.4560] |
| Cosmos 7B | [0.1129, 0.1602] | [0.2556, 0.3155] | [0.4314, 0.5623] |
| Cosmos 14B | [0.1689, 0.2254] | [0.2810, 0.3303] | [0.4147, 0.5076] |
| Hunyuan Video | [0.1108, 0.1558] | [0.2886, 0.3493] | [0.5129, 0.6097] |
| MAGI | [0.1126, 0.1621] | [0.2675, 0.3264] | [0.4401, 0.5545] |

Taken together, the bootstrap analysis demonstrates that the benchmark reliably distinguishes model performance, that prompt upsampling delivers stable improvements, and that difficulty stratification reflects meaningful variation in physical reasoning demands.

## J LIMITATIONS

While our primary focus is on open-source T2V models to ensure reproducibility and transparency, we also include a small number of proprietary models to broaden the scope of the evaluation. Further exploration of additional closed-source models remains a valuable direction for future research. Additionally, our use of LLMs for caption-based question answering may introduce occasional errors, as these models can produce inaccurate or hallucinated outputs. However, we mitigate this risk through cross–judge stability: replacing Gemini–Flash with DeepSeek yields nearly identical model rankings (Spearman $\rho=0.98$; see Sec. H.2), confirming that our evaluation signal reflects differences in the generated *videos*, not idiosyncrasies of a specific judge model.

A second limitation of our pipeline is that it is fully text-based: the judge LLM never observes the raw video and relies entirely on captioned descriptions. This introduces a dependency on caption quality, since omissions, ambiguities, or misinterpretations in the captions may propagate into the final judgment. We mitigate this through multi-perspective captioning (Sec. C.3) and cross–captioner stability analysis (Sec. H.3), which show that using AuroraCap, Qwen2.5-VL, or Qwen3-VL produces highly consistent scores.

## K    LLM USAGE

LLMs were used to polish the writing and improve the clarity and flow of the text. All final revisions were reviewed and edited by the authors.

