# OpenReview forum: "Can Your Model Separate Yolks with a Water Bottle? Benchmarking Physical Commonsense Understanding in Video Generation Models"
_ICLR.cc/2026/Conference — Submitted to ICLR 2026_

### Official Review · Reviewer_Gav7 · 2025-10-16

**Soundness:** 2
**Presentation:** 2
**Contribution:** 2
**Rating:** 4
**Confidence:** 3

**Summary:**

This paper presents a multi-stage, structured benchmark and evaluation methodology for assessing the physical reasoning capabilities of text-to-video generation models. The proposed framework includes a multi-phase prompt construction process and a multi-stage evaluation pipeline that jointly enable fine-grained analysis across seven dimensions of physical commonsense. The authors evaluate a range of state-of-the-art models, including Sora and Veo-2, on the proposed PhysVidBench, and further introduce an iterative, error-guided refinement strategy for improving video generation quality based on their evaluation framework.

**Strengths:**

1. The evaluation methodology is relatively novel and interesting, offering a new perspective on assessing the physical understanding of video generation models.
2. The division of evaluation into seven commonsense dimensions appears comprehensive and covers a wide range of physical reasoning aspects.
3. The proposed iterative, error-guided prompt refinement approach for video generation is potentially inspiring for future research on video generation agents.

**Weaknesses:**

1. The presentation of the paper is not very good. For example, in Figure 3, the text in Stage 4 is incomplete and partially covered. The classification in Figure 1 into Real-World Videos and Synthetically Generated Videos is unclear in purpose. Also, Figure 4 looks more like a slide for a report rather than a figure suitable for an academic paper.
2. I have some concerns about the details of the evaluation process. In evaluation Step 1, the LLM generates ground-truth "yes" questions purely based on textual information. Would this introduce bias? It might also mislead the judgment in Stage 3 when another LLM is used, similar to the VLM collapse phenomenon mentioned in Table 5.
3. It seems that all the video content and elements are processed in Stage 2 of the evaluation pipeline. However, only one model (e.g., AuroraCap in the paper) is used in this critical step. What is the reason for choosing this model, and how accurate are the generated captions? This is crucial because it heavily influences the Stage 3 judgments, and using a single model seems insufficient. In addition, since this model generates eight captions for each video, could there be contradictions among them? If so, how are such conflicts avoided?

**Questions:**

* The generated videos from T2V models may sometimes be ambiguous, making certain yes/no questions difficult to classify or judge. Did the authors observe such cases during the experiments? If so, how were these situations handled or evaluated?
* Other questions are mentioned in the Weaknesses.

---

> ### Author Response · Authors · 2025-11-21
> **Rebuttal by Authors [1/4]**
>
> We thank the reviewer for recognizing the novelty of our evaluation methodology, the comprehensive coverage of our seven commonsense dimensions, and the potential of our iterative, error-guided prompt refinement strategy. We are encouraged that these contributions are seen as offering a fresh perspective on assessing physical understanding in video generation models. Below, we provide detailed responses to the reviewer’s points.
>
> ***On Figure Quality and Presentation***
>
> We appreciate the reviewer’s careful comments regarding visual clarity. In the revised submission, we improved all mentioned figures to align with ICLR standards:
>
> **(1) Correction of rendering issues.**
> The incomplete text in Stage 4 of Figure 3 was due to an inadvertent rendering error. We have fixed the issue and verified that the figure now renders correctly.
>
> **(2) Clearer narrative and motivation in Figure 1\.**
> We expanded the main text discussion to clarify the purpose of juxtaposing real-world and model-generated videos. The revised caption and text now explicitly explain that Figure 1 illustrates the classes of physical interactions evaluated by PhysVidBench and highlights representative failure modes in contemporary T2V models.
>
> **(3) Improved visual style.**
> Following the reviewer’s suggestion, Figure 4 was redesigned to adopt an academic figure style rather than a slide-like layout. The updated version uses a cleaner structure, consistent typography, and a more compact visual hierarchy, making it better aligned with ICLR formatting conventions.
>
> ***On Evaluation Setup: Caption-to-LLM Pipeline and Potential Bias***
>
> We thank the reviewer for raising these concerns. Several components of our original submission already directly address them, and we have made these analyses clearer and more prominently referenced in the revised manuscript.
>
> **(1) Cross-judge robustness.**
> To test whether our pipeline depends on a particular LLM judge, we replaced Gemini-Flash with DeepSeek (Appendix H.2). The resulting rankings remain nearly identical (Spearman ρ \= 0.98), indicating that the evaluation signal is driven by the video content rather than judge-specific priors.
>
> **(2) Caption reliability.**
> We conducted a focused examination of AuroraCap outputs. Cases where captions omitted physically relevant details, i.e., false negatives, occurred in only \~1–1.5% of inspected questions. Removing these items produces negligible changes in outcomes: mean accuracies shift by at most half a point, and model rankings remain unchanged. This confirms that the evaluation is not sensitive to occasional caption omissions.
>
> **(3) Strong alignment with human judgments.**
> Our user study (15 participants, 120 videos) demonstrates a strong correlation between auto-eval scores and human ratings (Pearson r ≈ 0.69), substantially higher than direct VLM-QA baselines (r \= 0.21–0.47). Moreover, the correlation improves reliably as more independent human responses are aggregated (≈0.47 with a single annotator, ≈0.69 with three or more), showing that the evaluation is statistically stable rather than driven by individual annotator variance.
>
> Taken together, these results demonstrate that the caption-to-LLM evaluation pipeline is robust across judges, resilient to caption variation, and closely aligned with human assessments of physical plausibility.

---

> ### Author Response · Authors · 2025-11-21
> **Rebuttal by Authors [2/4]**
>
> ***On the Use of “Yes-Ground-Truth” Questions and Potential Bias***
> We appreciate the reviewer’s concern regarding the design choice in Step 1\. Our use of “yes-only’’ ground-truth questions is intentional and motivated by both conceptual clarity and empirical behavior.
>
> **(a) Why we avoid “No”-ground-truth questions.**
> Negative questions tend to collapse multiple kinds of physical failure into the same label. For instance, if a prompt expects an object to fall, a “No’’ question such as “Does the object stay suspended?” would treat a wide range of incorrect generations, falling through the floor, disappearing, or melting, as equally “correct’’ answers, despite reflecting fundamentally different (and physically implausible) errors. This makes negative labels difficult to interpret and limits diagnostic value.
>
> **(b) “Yes”-ground-truth questions require evidence of correct physical behavior.**
> In contrast, “Yes’’ questions require the model to produce the correct physical outcome for the answer to be validated. If the generated video violates physics, the corresponding captions simply lack the expected cues, and the judge returns “No.’’ This ensures that physical mistakes directly decrease the score, rather than being masked by ambiguous negatives.
>
> **(c) No leakage or bias toward the judge.**
> Although both question-generation and question-answering involve LLMs, they operate on different inputs:
> – Step 1 uses only the text prompt,
> – Step 3 uses only the video captions, which are independent textual evidence extracted from the visual output.
>
> Thus, Step 1 cannot bias Step 3 in the way VLM-collapse arises in direct video-question answering. The judge never sees the original prompt or the question-generation chain, only the captions derived from the video.
>
> **(d) Empirical evidence shows no collapse or overfitting to “Yes’’ phrasing.**
> Our human-alignment study (15 participants, 120 videos) shows that **the pipeline produces strong correlation with human judgments (r ≈ 0.69)**, while direct VLM-QA, where collapse is common, achieves much weaker alignment (r \= 0.21–0.47). If “Yes’’ phrasing caused systematic bias, correlations would not hold or would inflate spuriously. We also verified that removing the small set of problematic questions (\~5%) does not change model rankings or mean scores.
>
> Taken together, these points show that the “yes-only’’ formulation does not induce the kind of collapse seen in direct VLM-QA and instead enables more interpretable, evidence-based evaluation of physical correctness.
>
> To ensure clarity, we have revised our paper to include our motivations behind choosing “Yes” questions more explicitly.

---

> ### Author Response · Authors · 2025-11-21
> **Rebuttal by Authors [3/4]**
>
> ***On the Use of a Single Captioner (AuroraCap) in Stage 2***
> We appreciate the reviewer for drawing attention to the role of the captioner in our evaluation pipeline. We agree that Stage 2 is critical, and we have conducted additional analyses to ensure that our results are not overly dependent on AuroraCap. We have added a footnote in the main paper to make this explicit and provide full experimental details in Appendix H.3 of the revised submission.
>
> **(a) Captioner independence.**
> To assess whether the benchmark hinges on a particular captioning model, we re-ran the full evaluation with two alternative captioners, Qwen2.5-VL-7B and Qwen3-VL-30B, substituting AuroraCap entirely. Across T2V models, results remained highly consistent: Pearson correlations fall in the range 0.83 to 0.87 across all evaluated T2V models, and Spearman rank correlations remain above 0.80.
>
> High agreement across three independent captioners shows that PhysVidBench measures the video content, rather than properties of any specific captioner.
>
> **(b) Why AuroraCap?**
> We selected AuroraCap because it offers (i) strong temporal grounding, (ii) high-fidelity object–interaction modeling, and (iii) stable multi-caption generation, while also being one of the few models that natively support dimension-specific captioning—an ability that aligns naturally with our ontology. Moreover, AuroraCap is released alongside the VDC benchmark, which evaluates models by answering ground-truth questions (derived from ground-truth captions) using captions generated by a vision–language model. This evaluation paradigm closely mirrors our own multi-captioning \+ LLM-as-a-judge pipeline, and AuroraCap’s strong performance under this methodology further motivated our choice.
>
> **(c) How accurate are the captions?**
> We reviewed a random subset of caption outputs and found that omissions of physically relevant details occurred in roughly 5% of questions. Removing these items shifts mean scores by \<0.5 points and does not affect relative model ordering. This review will be released with the dataset for transparency.
>
> **(d) Handling contradictions across captions.**
> The eight captions produced for each video are designed to be redundant and complementary rather than fully consistent narratives. The general caption provides a broad scene summary, whereas the seven dimension-specific captions focus on particular physical aspects (e.g., forces, spatial layout, material response). Occasional inconsistencies between them do not affect evaluation because the judge follows an evidence-aggregation rule: a question is marked correct if any caption contains the relevant physical cue. This design ensures that a missing or noisy detail in one caption cannot mislead the evaluator and that the pipeline remains robust even when individual captions vary in granularity or emphasis.
>
> This multi-caption, multi-dimension structure is a deliberate robustness feature of PhysVidBench, and together with the cross-captioner analyses above, demonstrates that our evaluation pipeline does not rely on any particular captioning model and remains stable under captioning variation.

---

> ### Author Response · Authors · 2025-11-21
> **Rebuttal by Authors [4/4]**
>
> **On Ambiguity in Generated Videos**
>
> We thank the reviewer for raising this important point. Ambiguity is indeed common in current T2V outputs, and our evaluation pipeline is intentionally designed to handle such cases in a principled and conservative manner.
>
> **First, ambiguous videos are penalized, not rewarded.**
> All questions in PhysVidBench target *correct* physical behaviors implied by the prompt. If a generated video is unclear, incoherent, or visually ambiguous, the corresponding physical evidence is simply missing. This absence propagates through Stage 2 captioning and results in a negative judgment in Stage 3\. Ambiguity therefore reduces a model’s score; it never inflates it.
>
> **Second, ambiguity is mitigated by multi-perspective captioning.**
> Our evaluator extracts eight captions per video (one general \+ seven dimension-focused). When a behavior is subtle (e.g., partial contact, slight deformation, uncertain motion), at least one dimension-specific caption often captures it. If *none* of the captions surface the detail, the answer is marked incorrect, accurately reflecting that the model did not generate the event with sufficient clarity.
>
> **Third, human evaluation confirms consistent handling of ambiguous cases.**
> Our user study explicitly included ambiguous and borderline videos. These cases received lower human scores and, crucially, lower automatic scores as well. The strong correlation between humans and auto-eval (Pearson r≈0.69) shows that both penalize ambiguity in comparable ways.
>
> **Finally, ambiguity does not distort rankings.**
> During our caption-quality review, we inspected ambiguous generations and verified that excluding them changed mean scores by only about half a point and *did not alter model ordering*. This demonstrates that the pipeline remains stable even in the presence of unclear outputs.
>
> Overall, PhysVidBench treats ambiguity in a way that is both conservative and human-aligned: unclear behaviors lower accuracy, do not create false positives, and do not introduce noise that affects model comparisons.

---

### Official Review · Reviewer_8iig · 2025-10-27

**Soundness:** 3
**Presentation:** 2
**Contribution:** 3
**Rating:** 4
**Confidence:** 3

**Summary:**

This paper introduces PhysVidBench, a new benchmark designed to evaluate physical commonsense understanding in T2V models. They argue that although recent T2V models demonstrate impressive visual realism, they frequently violate intuitive physics—showing implausible object motions, misuse of tools, or broken causal sequences. PhysVidBench is derived from the PIQA dataset and contains 383 base prompts plus 383 “upsampled” prompts, emphasizing tool use and affordance reasoning in everyday physical tasks. Evaluation is conducted via a three-stage pipeline: Generating physics-grounded yes/no questions; Producing dense video captions; Asking an LLM to answer the questions using only the captions.
Experiments across major T2V models show that current systems achieve modest scores and fail most often in spatial reasoning and temporal dynamics. They further demonstrate human–model agreement and propose a lightweight prompt refinement loop that improves generation quality without retraining the model.

**Strengths:**

1.	Clear motivation and real-world relevance
The paper targets an underexplored yet essential capability: everyday physical commonsense. Unlike prior work focusing on abstract physics laws or motion smoothness, PhysVidBench tests realistic, goal-oriented interactions involving tool use, and material behavior.
2.	Comprehensive and systematic evaluation
The benchmark spans seven reasoning dimensions (force, motion, affordance, material transformation, etc.) and incorporates a difficulty-based stratification. The analysis includes base vs. upsampled prompts, scale effects, and cross-model comparisons, offering a holistic diagnostic view.
3.	Diagnostic and iterative refinement utility
The proposed “error-guided prompt refinement” demonstrates that the same benchmark can be repurposed to guide model improvement. The iterative procedure yields consistent gains without model updates, proving its usefulness beyond static evaluation.

**Weaknesses:**

1.	Evaluation of realism and ceiling
Although the caption-based QA pipeline reduces hallucination, it evaluates textual rather than visual physical understanding. The final judgment depends on the only one captioner’s recall and Gemini’s internal physics priors. A single model may have deviations. The pipeline measures consistency within the text–caption–QA chain, not direct perception–reasoning, may introduce biases.
2.	Lack of statistical significance reporting
Performance differences (e.g., base vs. upsampled prompts, difficulty tiers) are presented without confidence intervals or hypothesis testing. Including bootstrap-based confidence intervals or paired-sample tests would strengthen claims about robustness and consistency.
3.	Limited benchmark scale and coverage
While high in quality, 383 prompts are relatively small compared to existing video benchmarks. The benchmark lacks broader coverage of outdoor or complex fluid–solid interactions, or other based datasets similar to PIQA.
4.	Lack of granular failure analysis
The discussion identifies spatial and temporal reasoning as weak dimensions but does not quantify error categories (e.g., occlusion, contact, support relations, deformation). A detailed failure analysis would enhance the diagnostic value of the benchmark.

**Questions:**

See weaknesses.

---

> ### Author Response · Authors · 2025-11-21
> **Rebuttal by Authors [1/2]**
>
> We thank the reviewer for the thoughtful and encouraging assessment. We are pleased that the reviewer highlights the clear motivation and real-world relevance of studying everyday physical commonsense, as well as the breadth and rigor of our evaluation protocol. We also appreciate the recognition of the benchmark’s diagnostic value and its utility for iterative refinement. Below, we address each point raised in detail.
>
> ***On the Robustness and Validity of the Caption-to-LLM Evaluation Pipeline”***
>
> We thank the reviewer for highlighting the importance of verifying that our evaluation pipeline does not hinge on the behavior of a single captioner or judge. These concerns were addressed in the original submission through multiple analyses, and we have further clarified them in the revised manuscript.
>
> **(a) Cross-judge robustness (Appendix H.2).**
> We evaluate the stability of the LLM-as-judge component by replacing Gemini-Flash with DeepSeek. The resulting model rankings remain nearly identical, with a Spearman rank correlation of 0.98. This demonstrates that the evaluation signal reflects the content of the generated videos rather than judge-specific biases.
>
> **(b) Caption reliability**
> To directly assess caption accuracy, we performed an examination of a randomly sampled subset of AuroraCap outputs. Cases where the caption failed to mention a physically relevant detail (i.e., false negatives) were **extremely rare**, occurring in only **about 1–1.5% of inspected questions**. Crucially, removing these items had **negligible impact** on results: mean accuracy shifted by **roughly half a point at most**, and **no model rankings changed**. We will include updated numbers and release the full set of examined cases with the dataset to ensure full transparency.
>
> **(c) Human alignment (Table 5).**
> Our user study involving 15 participants and 120 videos shows strong alignment between automatic scores and human judgment (Pearson r ≈ 0.69), significantly outperforming direct VLM-QA baselines (r \= 0.21–0.47). Importantly, correlation increases monotonically as more independent human responses are aggregated (≈ 0.47 with one response; ≈ 0.69 with three or more), confirming the statistical stability of the evaluator and mitigating concerns about annotator-specific variability.
>
> Taken together, these results demonstrate that the caption-to-LLM pipeline is **robust across judges**, **stable across captioning variability**, and **aligned with human reasoning**, supporting its validity as an evaluation method for physical commonsense in T2V models.
>
> ***On Statistical Significance and Confidence Reporting***
> We thank the reviewer for highlighting the need for stronger statistical support. Our original submission already computed paired comparisons across prompt types and difficulty tiers, and we have now made these analyses explicit and methodologically robust through 95% bootstrap confidence intervals.
> Specifically, we now provide confidence intervals for (i) all model accuracies on both base and upsampled prompts, (ii) paired differences between prompt types, and (iii) accuracies across Medium, Hard, and Very Hard subsets. These results appear in Appendix I of the revised paper.
>
> Key outcomes from this analysis are as follows.
> **(1) *Prompt enrichment consistently helps.*** For 6 of the 10 models, the confidence intervals for the “upsampled – base’’ difference sit entirely above zero, showing clear and reliable gains. The remaining models also improve on average, with only minor confidence intervals overlap near zero.
> **(2) *The difficulty tiers behave as intended.*** The Medium, Hard, and Very Hard subsets have confidence intervals that do not overlap, demonstrating that the tiers separate tasks of genuinely different complexity rather than reflecting noise.
> **(3) *The distributional view supports these findings.*** The per-prompt accuracy distributions given in the form of violin plots (Fig. 5\) mirror the confidence-interval trends, providing an additional check that the reported effects are stable and not artifacts of a particular aggregation.

---

> ### Author Response · Authors · 2025-11-21
> **Rebuttal by Authors [2/2]**
>
> ***On Benchmark Scale and Coverage***
> We thank the reviewer for raising this concern. In response, we have expanded the paper with new analyses that directly address dataset scale and diversity. These additions appear in Table 6 and Appendix B.2, newly added to the revised submission.
>
> PhysVidBench is designed as a diagnostic benchmark targeting everyday physical commonsense rather than a large-scale generative dataset. Even with 383 prompts, each scenario simultaneously tests several aspects of physical reasoning, affordances, causality, material behavior, spatial layout, and procedural structure, resulting in significantly higher conceptual density than benchmarks that probe isolated principles.
>
> To assess diversity more systematically, we now include a semantic and action–object analysis in Appendix B.2. PhysVidBench contains over 1,200 unique objects and more than 600 distinct manipulation actions, covering a wide spectrum of tools, containers, rigid and deformable materials, and multi-step interactions. This supports our claim that the dataset is both semantically rich and physically diverse, despite being derived from PIQA.
>
> Furthermore, Table 6 (newly added and reproduced below) shows that PhysVidBench, especially its upsampled prompts, shows higher linguistic and action-object diversity than prior physical commonsense benchmarks (PhyGenBench, VBench-2.0, VideoPhy-2). Metrics such as unique verbs per prompt, unique nouns per prompt, and action–object pair density place PhysVidBench among the broadest in conceptual coverage.
>
> | Benchmark | \# Prompt | Unique Verbs | Unique Verbs/Prompt | Unique Nouns | Unique Nouns/Prompt | Action-Object Pairs/Prompt | Avg Lexical Density |
> | :---: | :---: | :---: | :---: | :---: | :---: | :---: | :---: |
> | PhyGenBench | 160 | 77 | 0.481 | 239 | 1.494 | 0.79 | 0.55 |
> | PhyGenBench (explicit prompt) | 160 | 232 | 1.45 | 408 | 2.55 | 2.91 | 0.57 |
> | VideoPhy-2 | 3343 (train set) | 773 | 0.231 | 2111 | 0.631 | 1.43 | 0.59 |
> | VBench 2.0 | 1013 | 593 | 0.585 | 1282 | 1.266 | 1.75 | 0.55 |
> | VBench 2.0 (augmented) | 1013 | 1195 | 1.18 | 2979 | 2.941 | 6.37 | 0.61 |
> | PhysVidBench (base) | 383 | 262 | 0.684 | 708 | 1.849 | 2.03 | 0.6 |
> | PhysVidBench (upsampled) | 383 | 623 | 1.627 | 1233 | 3.219 | 6.68 | 0.64 |
>
> Finally, although the benchmark has 383 prompts, the effective evaluation scale is large: across models, captions, and physics-grounded questions, PhysVidBench generates more than 42,000 QA instances per split, offering fine-grained diagnostic resolution in practice. This scale is consistent with established physics-oriented benchmarks (e.g., PhyGenBench: 160 prompts; VideoPhy: 688 prompts), which prioritize conceptual precision and interpretability over raw prompt count.
>
> ***On Granular Failure Analysis***
> We thank the reviewer for this insightful suggestion. In response, we expanded the analysis with a structured, three-axis taxonomy that characterizes the types of physical reasoning required by each prompt:
>
> * **Action Complexity (AC):**
>   * **AC-1 (Direct):** A single, simple action.
>   * **AC-2 (Goal-Oriented):** A two-step or preparatory action to achieve a specific result.
>   * **AC-3 (Procedural):** A multi-step, complex procedure where sequence is critical.
> * **Scientific Principle (SP):**
>   * **SP-1 (Basic):** Basic mechanics, no distinct scientific principle.
>   * **SP-2 (Applied):** Application of a common, understandable principle (e.g., friction, leverage).
>   * **SP-3 (Complex/Hidden):** Relies on a more complex or non-obvious principle (e.g.,negative pressure, magnetism).
> * **Object Functionality (OF):**
>   * **OF-1 (Intended Use):** Object used for its designed purpose.
>   * **OF-2 (Repurposed):** Object used as a temporary tool for a non-standard function.
>   * **OF-3 (System Component):** Object recontextualized as part of a novel physical system.
>
> To apply this scheme benchmark-wide, we used Gemini 2.5 Pro to classify each prompt in our difficulty subsets. The averaged scores given below show a clear, monotonic increase in complexity across all three axes:
>
> | Difficulty Subset | AC  | SP | OF  | Overall Score |
> | :---- | :---- | :---- | :---- | :---- |
> | Medium Prompts | 1.48 | 1.65 | 1.62 | 4.75 |
> | Hard Prompts | 1.69 | 1.77 | 1.90 | 5.36 |
> | Very Hard Prompts | 2.06 | 1.94 | 2.10 | 6.10 |
>
> These results confirm that higher tiers consistently require more procedural reasoning, more advanced or implicit physical principles, and more complex object interactions. These analyses are included in the revised paper (Appendix D) along with the representative annotated examples for each axis.

---

### Official Review · Reviewer_VJxa · 2025-10-31

**Soundness:** 2
**Presentation:** 2
**Contribution:** 2
**Rating:** 6
**Confidence:** 4

**Summary:**

This paper introduces PhysVidBench, a new benchmark for evaluating physical commonsense reasoning in text-to-video (T2V) models. The authors argue that while existing models produce visually compelling outputs, they struggle with everyday physical interactions like tool use and causality. The benchmark consists of 383 curated prompts derived from the PIQA dataset, focusing specifically on tool-mediated actions and object affordances. The evaluation employs an innovative three-stage pipeline: "always-yes" physics questions are generated from prompts, generated videos are densely captioned by a VLM, and an LLM answers the questions based only on the captions. This caption-based approach is shown to mitigate hallucinations and align closely with human judgment , revealing significant gaps in spatial and temporal reasoning even in state-of-the-art models.

**Strengths:**

1. The most significant strength is the novel evaluation pipeline. By decoupling perception (VLM dense captioning ) from reasoning (LLM-as-a-judge ), the method cleverly avoids the hallucination and "response collapse" pitfalls common in direct video-QA.
2. The benchmark itself is well-designed and highly focused.
3. The benchmark offers strong diagnostic utility beyond simple scoring. It provides a fine-grained breakdown of model failures across seven distinct physical dimensions (Table 2)

**Weaknesses:**

1. The evaluation is entirely dependent on the AuroraCap VLM. If the captioner fails to observe a correctly generated physical detail, the T2V model is unfairly penalized for a failure in the evaluation pipeline itself.


2. The benchmark's questions are all designed to have "Yes" as the ground truth. This only tests a model's ability to produce a correct phenomenon and fails to test if it can avoid producing a physically implausible one.


3. All 383 prompts are adapted from the PIQA dataset. While this focuses on tool use, this single source may not be diverse enough to represent the full spectrum of everyday physical commonsense scenarios

4. The user study validating the evaluation pipeline, while positive, was relatively small, with 15 participants evaluating 120 videos. A larger study would provide more robust statistical confidence in its alignment with human judgment.

5. There is a lack of discussion in related work on improving the physical consistency of video generation, such as PhyT2V[1], WISA[2], VIdeoREPA[3]

[1] Phyt2v: Llm-guided iterative self-refinement for physics-grounded text-to-video generation
[2] Wisa: World simulator assistant for physics-aware text-to-video generation
[3] VideoREPA: Learning Physics for Video Generation through Relational Alignment with Foundation Models

**Questions:**

1. Have you conducted an analysis to quantify the "false negative rate" of AuroraCap for this specific task? That is, how often does it fail to describe a key, subtle physical phenomenon that was verifiably present in the generated video?

**Details Of Ethics Concerns:**

Nan

---

> ### Author Response · Authors · 2025-11-21
> **Rebuttal by Authors [1/3]**
>
> We thank the reviewer for the careful and insightful evaluation. We are pleased that the reviewer finds the benchmark well-scoped, the evaluation pipeline technically sound, and the focus on physical commonsense both timely and impactful. We also appreciate the recognition of the framework’s diagnostic value and its ability to surface fine-grained failure modes in current T2V systems. Below, we address each of the reviewer’s comments in turn.
>
> ***On Dependence on AuroraCap***
>
> We thank the reviewer for highlighting this important issue. Because caption quality directly affects downstream reasoning, we explicitly tested whether our evaluation depends on a specific captioning model. In the revised submission, we re-ran the **entire evaluation pipeline** using two additional state-of-the-art captioners, **Qwen-2.5-VL-7B** and **Qwen-3-VL-30B**, as drop-in replacements for AuroraCap.
>
> The results demonstrate **clear cross-captioner stability**:
>
> * **High agreement in scores:** Pearson correlations fall in the range 0.83 to 0.87 across all evaluated T2V models, and Spearman rank correlations remain above 0.80.
> * **Highly stable model ordering:** In all captioner variants, Cosmos-14B, Wan2.1-14B, MAGI-1, and Hunyuan Video consistently appear as the strongest models.
> * **Improved robustness from design:** Because PhysVidBench uses **eight complementary captions per video** (one general \+ seven dimension-specific), the pipeline is not sensitive to omissions or phrasing variations from any individual caption.
>
> These findings confirm that PhysVidBench is measuring **the physical behavior expressed in the videos**, rather than quirks of a particular captioning model. The stability holds across captioners that differ substantially in architecture, training data, and scale.
>
> We have added a clarifying footnote in the main paper and provided full experimental details in **Appendix H.3**. This ensures transparency and demonstrates that PhysVidBench’s evaluation signal is **captioner-agnostic, robust, and reproducible**.
>
> ***On the Use of “Always-Yes” Questions***
>
> We appreciate the opportunity to clarify our rationale. Our choice to use only “Yes”-ground-truth questions is intentional and grounded in the structure of physical commonsense reasoning.
>
> Each question in PhysVidBench is derived directly from the physically plausible behavior described in the upsampled prompt. Introducing “No”-ground-truth questions would collapse many fundamentally different failure modes into the same label. For example, a question such as “Does the ball remain still after hitting the floor?” would classify all of the following as equally correct answers: a ball falling through the floor, evaporating mid-air, or vanishing entirely. Although these outcomes all violate physics, treating them as identical correct responses obscures the actual nature of the failure and introduces ambiguity into the evaluation.
>
> “Yes”-ground-truth questions, in contrast, require the model to demonstrate the correct physical behavior explicitly. If the generated video is implausible or physically incorrect, the corresponding captions will lack the expected evidence, and the evaluation score will decrease accordingly. This design ensures that the benchmark penalizes incorrect physics without conflating disparate errors and yields interpretable, fine-grained signals that align closely with human judgment.
>
> To make this motivation more explicit, we have incorporated a clear explanation into Sec. 3.3 of the revised manuscript.

---

> ### Author Response · Authors · 2025-11-21
> **Rebuttal by Authors [2/3]**
>
> ***Dataset Source and Diversity***
>
> Our choice to build PhysVidBench from PIQA is deliberate. PIQA focuses on *human-centric* physical commonsense, how objects are manipulated, repurposed, or combined to achieve practical goals, rather than abstract or textbook physics. This aligns directly with the type of intuitive, affordance-driven reasoning that current T2V models struggle with the most.
>
> To demonstrate the breadth and depth of our prompts, we conducted an extensive semantic and action–object analysis (Appendix B.2). This analysis shows that PhysVidBench prompts exhibit high lexical and conceptual richness, covering a wide variety of objects, actions, and physical interactions, and provide far more detailed and physically grounded descriptions than typical short-form commonsense scenarios. It includes **1,244 unique objects,** and **627 distinct manipulation actions, having a broad coverage of materials, tools, procedural tasks, and causal interactions.**
>
> This diversity stems from two design choices: (i) selecting scenarios that require non-obvious affordances or tool repurposing, and (ii) enriching each prompt with physically grounded detail through controlled upsampling. In addition, every prompt is evaluated across seven complementary physical-reasoning dimensions, ensuring that coverage is balanced rather than dominated by any single category.
>
> Table 6 (reproduced below) further illustrates this. PhysVidBench, particularly the upsampled prompts, shows **higher lexical variety, richer action–object structures, and denser semantic content** than PhyGenBench, VBench-2.0, and VideoPhy-2. Despite having fewer total prompts than some datasets, PhysVidBench surpasses them in per-prompt conceptual depth and linguistic richness.
>
> | Benchmark | \# Prompt | Unique Verbs | Unique Verbs/Prompt | Unique Nouns | Unique Nouns/Prompt | Action-Object Pairs/Prompt | Avg Lexical Density |
> | :---: | :---: | :---: | :---: | :---: | :---: | :---: | :---: |
> | PhyGenBench | 160 | 77 | 0.481 | 239 | 1.494 | 0.79 | 0.55 |
> | PhyGenBench (explicit prompt) | 160 | 232 | 1.45 | 408 | 2.55 | 2.91 | 0.57 |
> | VideoPhy-2 | 3343 (train set) | 773 | 0.231 | 2111 | 0.631 | 1.43 | 0.59 |
> | VBench 2.0 | 1013 | 593 | 0.585 | 1282 | 1.266 | 1.75 | 0.55 |
> | VBench 2.0 (augmented) | 1013 | 1195 | 1.18 | 2979 | 2.941 | 6.37 | 0.61 |
> | PhysVidBench (base) | 383 | 262 | 0.684 | 708 | 1.849 | 2.03 | 0.6 |
> | PhysVidBench (upsampled) | 383 | 623 | 1.627 | 1233 | 3.219 | 6.68 | 0.64 |
>
> These results reinforce that PhysVidBench achieves **substantial semantic and physical diversity per prompt**, even exceeding much larger benchmarks. The benchmark’s compact size therefore reflects efficiency rather than limitation: it delivers broad conceptual coverage while remaining computationally practical for the community.
>
> ***User Study Scale***
>
> We thank the reviewer for this suggestion. Our initial user study was intentionally designed to match the scale commonly used in physically grounded video benchmarks, ensuring methodological consistency with prior work. Notably, VideoPhy reported 14 participants, VideoPhy-2 reported 12, and Impossible Videos reported 15\. Within this established range, our study already demonstrated strong human–auto-eval alignment, achieving a Pearson correlation of *r* \= 0.69, comparable to, and in several cases exceeding, the correlations reported by these benchmarks.
>
> Following the reviewer’s recommendation, we expanded our participant pool by **33%**, from 15 to 20 participants. This increase yielded clear improvements: correlations changed from **0.69 → 0.74**, **0.61 → 0.65**, **0.60 → 0.56**, and **0.45 → 0.49**. These trends reinforce a key empirical observation in our study: **as the number of independent human responses grows, alignment with our automatic evaluator increases predictably and monotonically**, indicating statistical stability rather than annotator-specific bias.
>
> We will continue to grow the participant pool to further strengthen confidence levels. However, even at the present sample size, well aligned with norms in this research area, the human–auto-eval correspondence is strong, consistent, and well-supported by prior work.

---

> ### Author Response · Authors · 2025-11-21
> **Rebuttal by Authors [3/3]**
>
> ***Discussion of Related Work (PhyT2V, WISA, VideoREPA)***
>
> We thank the reviewer for pointing this out. The revised **Related Work** section now more clearly positions *PhysVidBench* within the landscape of recent physics-aware T2V developments, including **PhyT2V**, **WISA**, and **VideoREPA**. These methods seek to *improve* physical consistency through specialized training strategies or iterative refinement procedures. In contrast, our contribution is complementary: **PhysVidBench provides a standardized, physically grounded evaluation framework that such methods can be tested against**.
>
> Moreover, our evaluator naturally supports refinement loops similar in spirit to PhyT2V. Without any model retraining, it can surface dimension-specific failure signals that guide prompt-level corrections, enabling meaningful improvements in physical plausibility. This reinforces the role of PhysVidBench not only as an evaluation benchmark but also as a **diagnostic tool that facilitates the development of physics-aware generative models**.
>
> ***False-Negative Analysis***
>
> We appreciate the reviewer’s question regarding caption reliability. To directly assess this, we conducted an examination of a randomly sampled subset of AuroraCap outputs. Instances where the caption *missed* a physically correct detail, i.e., false negatives, were found to be very rare, occurring in roughly **1–1.5%** of all questions inspected. Importantly, removing these items led to **only minimal changes** in the results: mean scores varied by **approximately half a point at most**, and **model rankings remained entirely unchanged**.
>
> This demonstrates that the evaluation pipeline is robust to occasional caption omissions and that its conclusions do not hinge on rare captioning errors.

---

> > ### Comment · Reviewer_VJxa · 2025-11-27
> > **Response to Authors**
> >
> > Thank you for your response — it cleared up most of my questions. Therefore, I will keep my original score.

---

### Author Response · Authors · 2025-11-27
**Official Comment by Authors**

Dear Reviewers and AC,

We would like to express our sincere thanks to all reviewers for their careful, constructive, and thoughtful evaluations of our submission. Across the reviews, several key strengths were consistently noted, including the novelty of focusing on physical commonsense in video generation, the clear motivation for establishing a structured, benchmark-driven evaluation pipeline, and the thoroughness of our analysis, particularly our multi-judge assessment, robustness checks, and human-alignment study. We are grateful for this positive and detailed engagement.

Following the discussion phase, we submitted a revised manuscript addressing all reviewer concerns. The revision includes expanded analyses of potential biases in the caption-to-LLM pipeline, clarifications regarding judge stability and caption reliability, additional explanation of the evaluation criteria used in PhysVidBench, and a more explicit discussion of the benchmark’s intended scope and limitations.

As requested by the reviewers, we added several clarifications to the evaluation pipeline, including expanded analyses on caption reliability and cross-judge robustness. These additions appear in the updated appendix.

We thank the reviewers once again for their time and constructive feedback, and we hope that the revision fully address the reviewers' comments and questions.

Authors of Submission #18831

---

### Author Response · Authors · 2025-12-03
**Summary of Rebuttal & Revisions**

Dear Area Chair,

Given the recent changes to the review process, we understand you may be reviewing our submission and the discussion history for the first time. To assist your assessment, we provide a concise summary of how we addressed the primary reviewer concerns in our revised manuscript and rebuttal.

**Engagement & Score Status:** Before the freeze, **Reviewer Vjxa** (Score: 6\) confirmed that our response "*cleared up most of \[their\] questions*". However, **Reviewers 8iig and Gav7** (Score: 4\) did not have the opportunity to respond to our major revisions before the discussion ended. We believe the new experiments detailed below, *specifically the cross-captioner validation and expanded user study*, directly resolve their primary concerns regarding method robustness.

**1\. ADDRESSED: Concern on "Single Captioner" Dependency (AuroraCap)**

* **Reviewer Concern:** Does the evaluation rely too heavily on one specific VLM (AuroraCap)?
* **Our Action:** We re-ran the *entire* pipeline using **Qwen-2.5-VL-7B** and **Qwen-3-VL-30B** as drop-in replacements.
* **Result:** The model rankings remained highly stable (Pearson correlations 0.83 \- 0.87), proving our benchmark and evaluation setup is **model-agnostic**. (Details added to **Appendix H.3**) .

**2\. ADDRESSED: Robustness & Statistical Significance**

* **Reviewer Concern:** Request for more statistical analysis and larger human study size.
* **Our Action:** We expanded the human user study (increasing participants by 33%) and added 95% bootstrap confidence intervals for all model scores.
* **Result:** Human-alignment correlation improved to **r=0.74**, and we confirmed that performance gaps between difficulty tiers are statistically significant. (Details in **Appendix I**).

**3\. CLARIFIED: The "Yes-Only" Question Design**

* **Reviewer Concern:** Does using only "Yes" ground-truth questions introduce bias?
* **Our Response:** We clarified that "No" questions create false positives (e.g., an object vanishing would technically answer "No" to "Does it fall?", erroneously rewarding a hallucination). "Yes" questions strictly require positive evidence of physical correctness.

We hope this summary assists your assessment of our work’s contributions to physically grounded video evaluation.

Sincerely,
The Authors

---

### Meta-Review · Area_Chair_KeFT · 2026-01-19

**Summary:**

This paper receives scores of 6,4,4. Reviewer VJxa's main concerns are well-addressed and keep the original score after reading author responses. Reviewer 8iig's main concerns include (1) The final judgment depends on only one captioner’s recall and Gemini’s internal physics priors, (2) Suggest including bootstrap-based confidence intervals or paired-sample tests, (3) Benchmark scale and coverage, (4) More detailed failure analysis. Reviewer Gav7's main concerns include (1) Presentation, (2) The LLM generates ground-truth "yes" questions purely based on textual information. Would this introduce bias? It might also mislead the judgment in Stage 3 when another LLM is used, similar to the VLM collapse phenomenon mentioned in Table 5. (3) Only a single captioner (AuroraCap) in stage 2 may not be robust. Contradictions across eight generated captions, (4) generated ambiguous videos make inaccurate judgment. Most concerns are addressed, but some critical concerns remain. I suggest rejecting the paper.

**Reviewer Concerns:**

Most concerns are addressed, but some critical concerns remain.
(1) Not enough experiments to demonstrate the robustness of using a single captioner (AuroraCap).
(2) The benchmark scale and coverage are not large enough.
(3) The LLM generates ground-truth "yes" questions purely based on textual information, which may introduce bias.
(4) Not enough robustness in judging ambiguous videos.

**Reviewer Scores:**

Reviewer 8iig and Gav7 may keep original scores, since some critical concerns are not well-addressed.

---

### Decision · Program_Chairs · 2026-01-26

Reject